# The Sea State CCI dataset v1 : towards a Sea State Climate Data Record based on satellite observations

Guillaume Dodet[1], Jean-François Piollé[1], Yves Quilfen[1], Saleh Abdalla[2], Mickaël Accensi[1], Fabrice Ardhuin[1], Ellis Ash[3], Jean-Raymond Bidlot[2], Christine Gommenginger[4], Gwendal Marechal[1], Marcello Passaro[5], Graham Quartly[6], Justin Stopa[7], Ben Timmermans[4], Ian Young[8], Paolo Cipollini[9], and Craig Donlon[10]

[1]Univ Brest, Ifremer, CNRS, IRD, LOPS, 29280 Plouzané,France
[2]European Centre for Medium-range Weather Forecasts, Reading, United Kingdom
[3]Satellite Oceanographic Consultants (SatOC), Coach House Farm, New Mills, Derbyshire, UK
[4]National Oceanography Centre (NOC), European Way, Southampton SO14 3ZH, United Kingdom
[5]Deutsches Geodätisches Forschungsinstitut der Technischen Universität München (DGFI-TUM), Arcisstrasse 21, 80333, München, Germany
[6]Plymouth Marine Laboratory (PML), Prospect Place, The Hoe, Plymouth, Devon, PL1 3DH, UK
[7]Department of Ocean Resources and Engineering, School of Ocean and Earth Science and Technology, University of Hawaii at Manoa, Honolulu, HI, USA
[8]Department of Infrastructure Engineering, University of Melbourne, Melbourne, Australia
[9]Telespazio VEGA UK for ESA Climate Office, ECSAT, Fermi Avenue, Harwell Campus Didcot, OX11 0FD, United Kingdom
[10]ESA Climate Office, ECSAT, Fermi Avenue, Harwell Campus Didcot, OX11 0FD, United Kingdom

**Correspondence:** Guillaume Dodet (guillaume.dodet@ifremer.fr)

**Abstract.** Sea state data are of major importance for climate studies, marine engineering, safety at sea, and coastal management. However, long-term sea state datasets are sparse and not always consistent, and sea state data users still mostly rely on numerical wave models for research and engineering applications. Facing the urgent need for a sea state Climate Data Record, the Global Climate Observing System has listed "Sea State" as an Essential Climate Variable (ECV), fostering the launch in 2018 of the Sea State Climate Change Initiative (CCI). The CCI is a program of the European Space Agency, whose objective is to realize the full potential of global Earth Observation archives established by ESA and its member states in order to contribute to the ECV database. This paper presents the implementation of the first release of the Sea State CCI dataset, the implementation and benefits of a high-level denoising method, its validation against in-situ measurements and numerical model outputs, and the future developments considered within the Sea State CCI project. The Sea State CCI dataset v1 is freely available on the ESA CCI website (http://cci.esa.int/data) at ftp://anon-ftp.ceda.ac.uk/neodc/esacci/sea_state/data/v1.1_release/. Three products are available: a multi-mission along-track L2P product (http://dx.doi.org/10.5285/f91cd3ee7b6243d5b7d41b9beaf397e1, Piollé et al., 2020a), a daily merged multi mission along-track L3 product (http://dx.doi.org/10.5285/3ef6a5a66e9947d39b356251909dc12b, Piollé et al., 2020b) and a multi-mission monthly gridded L4 product (http://dx.doi.org/10.5285/47140d618dcc40309e1edbca7e773478, Piollé et al., 2020c).

# 1 Introduction

Sea state, i.e. the description of wind sea and swell conditions at sea in terms of spectral or bulk wave parameters, is a key component of the coupling between the ocean and the atmosphere, the coasts and the sea ice. In the open ocean, wind-generated waves increase the sea surface roughness and enhance the air-sea momentum transfer through the modification of the wind stress (Edson et al., 2013). Wave breaking contributes to the mixing of the ocean upper layer (Babanin and Haus, 2009) and releases of sea spray aerosol into the atmosphere (Monahan et al., 1986). At the coast, waves are refracted by the shallow bathymetry and the tidal currents, they shoal over the shoreface and transfer energy to higher and lower harmonics through nonlinear interactions (Longuet-Higgins and Stewart, 1962). They eventually break in the surf zone, increasing the water level, generating strong currents and stirring large quantities of sediments at the break point (Thornton et al., 1996). All these wave-induced processes contribute to rapid coastal erosion (Masselink et al., 2016), dune breaching (Kraus and Wamsley, 2003) and/or low-lying land overwash during extreme storm events. In the high latitudes, waves interact with the sea ice by modifying its mechanical properties, through the fragmentation of the ice floes of the marginal ice zone into smaller pieces, or through the push of the ice in the direction of the wave propagation (Stopa et al., 2018). Given that increased greenhouse gas emission caused by anthropic activities has a strong impact on the Earth's climate, which translates into the modification of the atmospheric circulation, the acceleration of sea level rise and the rapid decay of Arctic sea ice, significant changes in future sea state conditions and the above-mentioned coupling mechanisms are expected (see e.g. Thomson and Rogers, 2014; Idier et al., 2019; Reguero et al., 2019).

Nowadays, long-term records of wave parameters are provided by Voluntary Observing Ships along the major maritime routes (Gulev and Grigorieva, 2004), by in-situ wave buoy networks, mostly located along the US, European, Japanese and Australian coastlines, and by satellite altimeter measurements (Ribal and Young, 2019). While altimeter-based datasets are providing the (almost) global coverage necessary to understand the large-scale variability of sea states and their interactions with the other components of the Earth's climate, they still suffer from several limitations: 1) the main sea state parameter computed from radar altimeter echoes is the significant wave height, yet other spectral parameters such as the wave period and directions are key for some applications, e.g. coastal impact (Dodet et al., 2019); 2) altimeter measurements cover the last 34 years only (starting with GEOSAT in 1985 with a data gap between 1990 and 1991), which is still relatively short to extract robust trend information out of the strong multi-annual fluctuations of the significant wave height; 3) the sparse altimeter sampling pattern and the changing number of in-orbit altimeter missions cause undersampling errors which bias the long-term statistics, in particular for extreme conditions (Jiang, 2020); 4) altimeter missions need to be accurately cross-calibrated to deliver consistent long-term time-series, this is particularly true when instruments operating in different modes are merged in a single product (Timmermans et al., 2020); 5) altimeter measurements are contaminated by different sources of noise, which prevent a proper representation of SWH variability at scale lower than 100 km (Ardhuin et al., 2017). In the last 20 years, several research groups have contributed to the development of long-term calibrated altimeter databases (Queffeulou, 2004; Zieger et al., 2009; Ribal and Young, 2019), and some of these datasets have been used to compute the significant wave height trends over the last decades. In a recent study, Young and Ribal (2019) estimated trends in SWH ranging from -1 to +1 cm

year$^{-1}$, depicting a large regional variability with negative trends mostly located in the Pacific Ocean. These results, and the dataset they are based on (Ribal and Young, 2019), represent a milestone in the characterization of sea state decadal variability, however, new developments are necessary to verify these findings and extend the potential of satellite sea state observations.

Aware of the increasing need for accurate, robust and consistent long term sea state data required by the climate science community (Ardhuin et al., 2019), the Global Climate Observing System (GCOS) has listed "Sea State" as an Essential Climate Variable (ECV). ECVs are geophysical records generated from systematic Earth Observations in support of the international frameworks and policies such as the United Nations Framework Convention on Climate Change (UNFCCC) and the Intergovernmental Panel on Climate Change (IPCC). The Climate Change Initiative (CCI) program, launched by the European Space Agency in 2010, has already contributed to the production of new Climate Data Records (CDR) associated with ECVs, such as aerosol (Popp et al., 2016) or sea ice concentration (Lavergne et al., 2019). In this context, the Sea State CCI project was kicked off in 2018 in order to produce a CDR for the new ECV "Sea State". This paper presents the first dataset released in the context of the Sea State CCI project.

The next section of this paper describes the altimeter missions that have been considered for the Sea State CCI dataset v1, and the in-situ and model data that have been used to compare against the altimeter data. Section 3 describes the main processing steps (namely, data editing, inter-calibration and denoising) implemented within the Sea State CCI production system. Section 4 presents the results of the comparison against in-situ measurements and model outputs. Section 5 presents two applications of the Sea State CCI dataset v1 at global and regional scales. Finally, Section 6 discusses the current status of the Sea State CCI datase v1, the main limitations of the data and the perspectives for the future releases of this dataset.

## 2 Data and Methods

### 2.1 Altimeter data

The altimeter data used in the Sea State CCI dataset v1 come from multiple missions spanning from 1991 to 2018. Although many spaceborne radar altimeters are bi-frequency for atmospheric corrections (Ku-C or Ku-S), only measurements in Ku band were used for consistency reasons, being available for all missions except SARAL/AltiKa (Ka band). Table 1 provides the list of missions used in the Sea State CCI dataset v1, together with the input product and version used, and their orbital properties (note that some cycle changes occurred in the course of some missions for limited period of times : these are not listed here for clarity but the corresponding measurements were included in the Sea State CCI dataset v1). Not surprisingly, the list of altimeter data sources is very similar to that used by the Sea Level CCI (Fig1a in Quartly et al., 2017), except that project could not utilise the instruments in very long repeat cycles.

### 2.2 In-situ measurements

The in-situ data used to validate the Sea State CCI dataset v1 were gathered by ECMWF (Figure 1). Most of the data came from the operational archive from ECMWF, where all data distributed via the Global Telecommunication System (GTS) are kept.

**Table 1.** Characteristics of altimeter missions used for the Sea State CCI dataset v1

| Mission | Instrument | Band | Covered period | Repeat period (days) | Altitude (km) | Inclination (°) | Source product |
|---------|-----------|------|----------------|----------------------|---------------|-----------------|----------------|
| ERS-1 | RA | Ku | 1991-2000 | 35 | 785 | 98.52 | OPR [ESA/F-PAF] |
| TOPEX | NRA | Ku | 1992-2006 | 10 | 1336 | 66 | MGDR [CNES] |
| ERS-2 | RA | Ku | 1995-2011 | 35 | 785 | 98.52 | OPR [ESA/F-PAF] |
| GFO | GFO-RA | Ku | 1998-2008 | 17 | 800 | 108 | GDR/POE [NOAA] |
| JASON-1 | Poseidon-2 | Ku | 2001-2013 | 10 | 1336 | 66 | GDR vE [AVISO] |
| ENVISAT | RA-2 | Ku | 2002-2012 | 35 | 799 | 98.55 | GDR v2.1 [ESA/F-PAC] |
| JASON-2 | Poseidon-3 | Ku | 2008-2019 | 10 | 1336 | 66 | GDR vD [AVISO] |
| CRYOSAT-2 | SIRAL | Ku | 2010-Ongoing | 369 | 717 | 92 | IGDR [NOAA] |
| SARAL | AltiKa | Ka | 2013-Ongoing | 35 | 785 | 98.55 | GDR [AVISO] |
| JASON-3 | Poseidon-3B | Ku | 2016-Ongoing | 10 | 1336 | 66 | GDR vD [AVISO] |

Data from moored buoys and fixed platforms were extracted. These data are usually reported hourly (or less frequently). The bulk of the data comes from moored buoys, with the exception of data from operating platforms in the North and Norwegian Seas and the Gulf of Mexico. The main data providers are the US, via the National Data Buoy Center (NDBC) and Scripps, Canada, the UK, France, Ireland, Norway, Iceland, Germany, Spain, Brazil, South Korea, and India. This dataset was supple-
5   mented by buoy data obtained from the websites from the UK Centre for Environment, Fisheries and Aquaculture Science (CEFAS) and the Faeroe Islands network. In addition, buoy data from Denmark, New-Zealand and Japan obtained as part of ECMWF wave forecast validation project were also used. A basic quality control was applied to each hourly time series for each location to remove spurious outliers.

Wave in-situ measurements were compared to altimeter data at every altimeter-in-situ match-up. An altimeter-in-situ match-
10  up occurs each time the altimeter ground track is less than 50 km from a in-situ location and the in-situ measurement is available within 30-min (following Queffeulou, 2004). For each match-up, the altimeter SWH is averaged over the along-track records lying within a 50-km-radius-circle centered on the in-situ location. The in-situ time-series are filtered with a 2-hr moving window and are then interpolated on the satellite overpass time. The metrics used for validations are the bias, the root mean

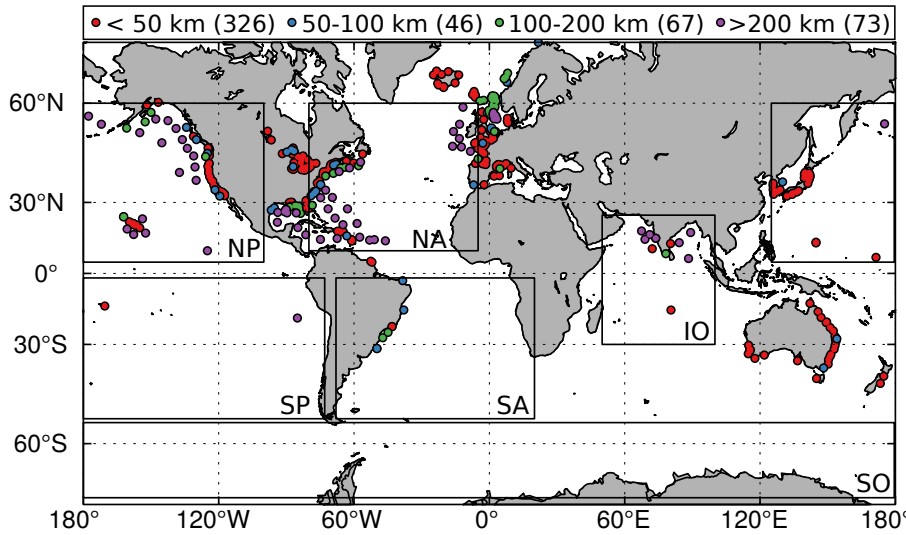

**Figure 1.** Global maps of the wave in-situ data used to validate the Sea State CCI dataset v1. Red circles indicate stations less than 50 km from the coast, blue circles indicate stations between 50-100 km from the coast, green circles indicate stations between 100-200 km from the coast, pink circles indicate stations more than 200 km from the coast. The number of stations is given within brackets. Black boxes indicate basins extensions used for the regional validation of the dataset.

square error (RMSE), the normalized RMSE (NRMSE), the scatter index (SI) and the coefficient of determination ($R^2$).

$$bias = \frac{1}{N} \sum X_{alti} - X_{ref}$$

$$RMSE = \sqrt{\frac{1}{N} \sum (X_{alti} - X_{ref})^2}$$

$$NRMSE = \sqrt{\frac{\sum (X_{alti} - X_{ref})^2}{\sum X_{ref}^2}}$$

$$SI = \sqrt{\frac{\sum (X_{alti} - \overline{X_{alti}})^2 - (X_{ref} - \overline{X_{ref}})^2}{\sum X_{ref}^2}}$$

$$R = \frac{\sum (X_{alti} - \overline{X_{alti}})(X_{ref} - \overline{X_{ref}})}{\sqrt{\sum (X_{alti} - \overline{X_{alti}})^2} \sqrt{\sum (X_{ref} - \overline{X_{ref}})^2}}$$

where $X_{alti}$ is the significant wave height recorded by the altimeter and $X_{ref}$ is the significant wave height recorded by the the wave buoy or the model (as mentioned in the next section). Comparisons between altimeter data and in-situ measurements showed much better agreement when coastal buoys (<200 km) were discarded from the analysis. This can be seen, for instance, on the scatter diagram and error metrics computed between SARAL and in-situ SWH measurements during the year 2017 (Figure 2), when all wave buoys are considered (left panel), and when only offshore wave buoys 200 km away from the coast are considered (right panel). Poorer performances in the comparison with coastal buoys have at least three reasons: firstly, land shading and refraction can modify SWH at much shorter distances than in the open ocean, affecting the validity of the 50-

km-radius assumption and jeopardising the number of sites that can be effectively used for the comparison; secondly, coastal backscatter inhomogeneities in the satellite footprint affects the retrievals particularly in the last 20 km from the coastline (see section 6.2); finally, the stronger variability of the wave field in the coastal zone due to tidal currents, bathymetric refraction, coastal wind in-homogeneity invalidates the assumption of wave field homogeneity within the altimeter footprint.

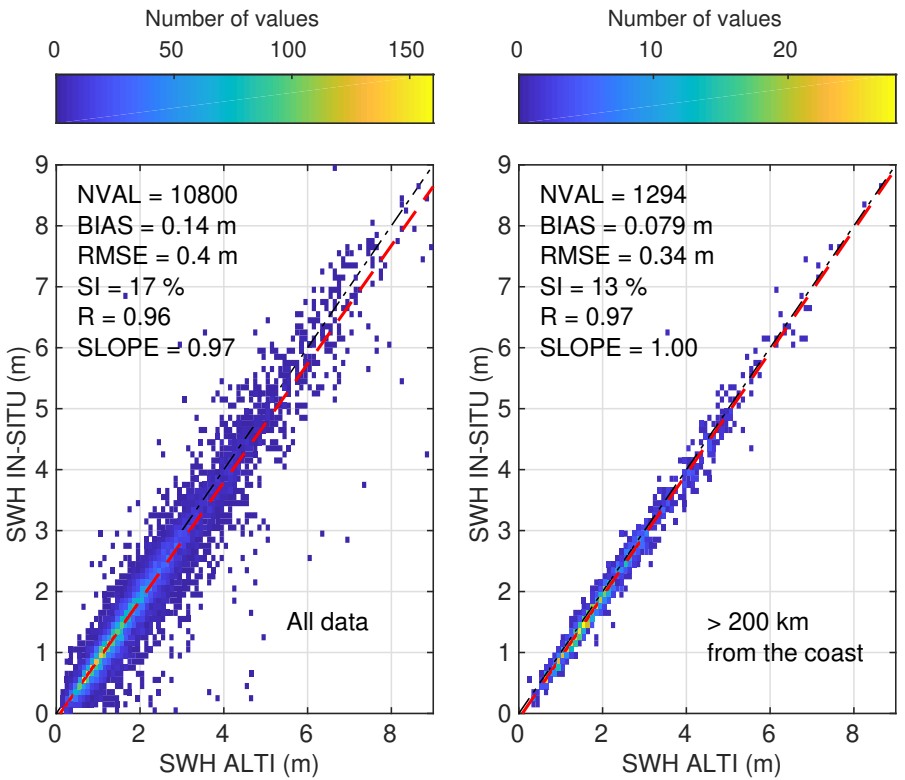

**Figure 2.** Comparison between SARAL and wave in-situ SWH measurements during year 2017, when all in-situ sites are considered (left) and when only locations 200 km away from the coast are considered (right).

### 2.3 Numerical wave model

The wave hindcast used to compare model results with altimeter data were produced with the spectral wave model WAVE-WATCH III © (WW3, The WAVEWATCH III Development Group, 2016). The model is forced by wind fields from the ERA5 reanalysis (Hersbach et al., 2018), by geostrophic and Ekman current components from Globcurrent products (Rio et al., 2014), with an ice mask applied from SSMI radiometer (CERSAT) and iceberg distribution from Altiberg (Tournadre et al., 2016). The coverage is global and extends from 78°S to 80°N at 0.5° resolution with a spectral discretization of 24 directions and 36 frequencies with lowest frequency at 0.0339 Hz. Output fields are generated at 3-hourly intervals. The WW3 version used is based on github NOAA-EMC stable released from June 27, 2019. The model parameterization is based on Rascle and Ard-

huin (2013) (T471) with following tuning for the wave growth : BETAMAX=1.65, SWELLF7 = $4.14 \times 10^5$ and for the strong wind intensification : WCOR1=23., WCOR2=1.08. Modelled SWH values are linearly interpolated along the satellite ground track and statistical errors (bias, RMSE, NRMSE, SI, $R^2$) are then computed. Statistics are only computed on measurements considered as good, based on the quality level flag defined in Section 3.1.

## 3 Processing of altimeter data

The Sea State CCI dataset v1 products are inherited from the GlobWave project (2009-2012) building on the experience and methodology developed within this project. It extends and improves the GlobWave products, which were a post-processing of existing L2 altimeter agency products with additional filtering, corrections and variables. Three kinds of products are delivered in the Sea State CCI dataset v1:

- L2P : Along-track products separated per satellite and half-orbit (pass) or full orbit (depending on the input product used), including all measurements with flags, corrections and extra parameters from other sources. These are expert products with rich content and no data loss (Piollé et al., 2020a);

- L3 : Edited merged daily products, derived from the L2P and retaining only valid and good quality measurements from all altimeters over one day (one daily file), with simplified content (only a few key parameters). This is close to what is delivered in near real time by, for instance, the Copernicus Marine Environment Monitoring Service (Piollé et al., 2020b);

- L4 : Statistical gridded products, also derived from the L2P and averaging valid and good measurements from all available altimeters over a fixed resolution grid (1°x1°) on a monthly basis. These products are meant for statistics and visualization (such as the CCI toolbox, http://climatetoolbox.io/) (Piollé et al., 2020c).

The following sections provide more details on the processing steps of L2P products, from which L3 and L4 are derived.

### 3.1 Data editing

This first step consists in the identification of bad or suspect measurements, in order to build a quality level flag (*swh_quality*) providing users with a way to only retain the valid measurements in their analysis. This is achieved through a series of tests applied to each measurement, the result of which are summarized into an additional rejection flag (*swh_rejection_flags*), where each bit documents a specific test's failure or success. Table 2 lists the four levels of the variable *swh_quality*.

When SWH measurements were rejected as bad, the reason (quality test) for which they were rejected is reported in the related *swh_rejection_flags* variable. The eight rejection flags are the following:

- **not_water**: The surface type is not water. It may be land, or continental ice. We try to keep lake and inner seas measurements (when the discrimination is possible from the GDR information). This test only uses the internal flags provided in the input product by the producer.

**Table 2.** Quality levels defined for Sea State CCI dataset v1

| Value | Meaning | Description |
|---|---|---|
| 0 | undefined | the measurement value is not defined or relevant (missing value, etc...), no quality check was applied. |
| 1 | bad | the measurement was qualified as not usable after quality check. |
| 2 | acceptable | the measurement may be usable for specific applications only or the quality check could not fully assess if it is a bad or good value (suspect). |
| 3 | good | the measurement is usable. |

- **sea_ice**: The measurement has possible ice contamination. The sea ice fraction is taken from an external source (such as the Sea Ice CCI microwave based daily maps). Sea ice contamination is defined as areas where the sea ice fraction is greater than a minimal threshold (corresponding to 10% of ice in the current configuration). SWH measurements where the sea ice fraction is greater than 0% but lower than 10% are classified as *acceptable*.

- **swh_validity**: The SWH measurements were considered as invalid (for instance because of the possible range or some internal flag provided in the original product used as input).

- **sigma0_validity**: The sigma0 measurements were considered as invalid for water surface type.

- **waveform_validity**: The measurements were considered as invalid as there are indications of unsuitable waveforms (as indicated in some internal flag provided in the original product used as input) for a proper SWH calculation.

- **ssh_validity**: The SWH measurements were considered as invalid as there were issues on SSH (as indicated in some internal flag provided in the original product used as input) which was considered as an indication of problematic quality for SWH too.

- **swh_rms_outlier**: The root mean square deviation of the 20 Hz SWH measurements exceeds a certain threshold, which depends on SWH and is computed following Sepulveda et al. (2015)

- **swh_outlier**: The measurements were considered as invalid when performing the SWH outlier test: this test considers all the measurements within a 100-km window centered on the screened measurement; measurements that deviate from the 100-km mean (excluding the two most extreme values in the mean calculation) by more than five standard deviations or by more than five meters are discarded. These empirical thresholds were defined through careful visual examination of the data. This step is iterated three times over the same window.

The editing criteria which leads to setting the SWH quality level and rejection flags are specific to each mission and are detailed in the Sea State CCI dataset product user guide (available on the project's website: http://cci.esa.int/seastate).

### 3.2 Cross-calibration

The Sea State CCI project builds on the GlobWave project, for which SWH altimeter measurements over the period 1985-2016 were carefully calibrated against in-situ data (GlobWaveTeam, 2013). In the Sea State CCI dataset v1, three additional altimeter missions, namely JASON-3, CRYOSAT-2 (Low Resolution Mode) and SARAL, have been included and we describe here the methodology used to cross-calibrate these SWH records against a common reference dataset. Moreover, a new version (version E) of the JASON-1 GDR has been released since the GlobWave project and the calibration formula derived for JASON-1 has also been updated. According to the GlobWave Annual Quality Control Report (GlobWaveTeam, 2012), there is no specific quality problem in JASON-2 and the variability in terms of data quality is lower than for JASON-1 and ENVISAT. Therefore, the calibrations of JASON-1, JASON-3, CRYOSAT-2 and SARAL are performed against the JASON-2 data, as calibrated by Queffeulou and Croizé-Fillon (2017). Altimeter SWH cross-calibration is carried out by comparing SWH measurements at cross-over locations between the altimeter to be calibrated and the reference mission JASON-2. A cross-over data pair is defined each time the two satellite ground tracks intersect within a 60-min time window (Fig. 3). In order to attenuate the impact of along-track noise (instrumental and retracking-induced noise) in the comparison, SWH is averaged along $n$ consecutive measurements 25-km apart of the intersection points ($7 \leq n \leq 9$ depending on altimeter orbital velocity, shown as blue and red dots on Figure 3). SWH at cross-over locations are then compared to estimate the calibration formula. Visual assessment of JASON-1, JASON-3 and SARAL SWH measurements against JASON-2 calibrated SWH measurements indicate a linear relationship between these missions (Figures 4, 5 and 6) and linear calibration formula are obtained by fitting a least-square regression line through the SWH data. Note that the fitting was only applied for SWH values larger than 1 m. Below this value, the linearity of the relationship is lost, mostly due to differences in the instrumental correction applied to account for the fact that the point target response in the model used to is approximated by a Gaussian function (Thibaut et al., 2010). Moreover, it is known that SWH retrieval at low sea states and particularly below 0.75 m is less accurate and noisier due the inadequate sampling of the signal (Smith and Scharroo, 2015). For CRYOSAT-2 the relationship is no longer linear (Figure 7) and we use a second-order polynomial function to correct this mission. In order to avoid discontinuous and unrealistic corrections at high sea state, we apply this second-order polynomial corrections until an upper threshold, corresponding to the SWH values at which the polynomial intersects the zero residual y-axis (in this case 7.67 m). Table 3 lists the equations used to calibrate the altimeter SWH measurements in the Sea State CCI dataset v1.

### 3.3 Data denoising

Altimeter measurements are characterized by a low signal-to-noise ratio (SNR) at spatial scales below about 100 km, blurring geophysical signal variabilities in this scale range, such as those resulting from wave-current interactions. The use of altimeter data therefore often requires preliminary noise filtering, and low-pass or smoothing filters are frequently applied. Such operation quite systematically results in the loss of small-scale (< 100 km) geophysical information or in the creation of artifacts in the geophysical variability analyzed (e.g. spectral ringing), and requires setting of a cut-off wavelength or a filter window length that is difficult to determine adequately for a global data set. As for approaches that infer a correction to eliminate

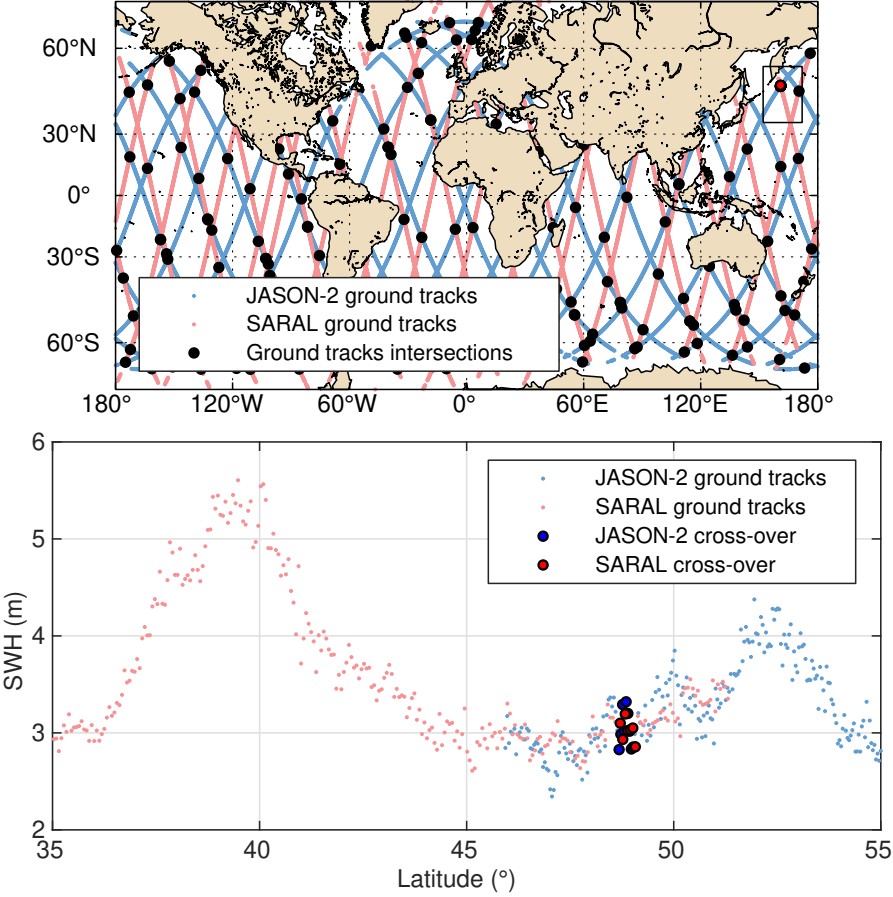

**Figure 3.** [Top panel] JASON-2 (light blue) and SARAL (light red) ground tracks on March 31 2018 with ground track intersection shown with black circles. [Bottom panel] Along-track data and cross-over highlighted in the top panel.

correlated errors from other aspects of the waveform data (Quartly, 2019; Tran et al., 2019), it also leaves a substantial amount of low- and medium-frequency noise in the data. To overcome these difficulties, an adaptive noise elimination method is used, based on the non-parametric Empirical Mode Decomposition (EMD) method developed to analyze non-stationary and non-linear signals (Huang et al., 1998). EMD is a scale decomposition into a limited number of amplitude and frequency modulated

5   functions (AM/FM) - called Intrinsic Mode Functions (IMF) - among which the Gaussian noise distribution is predictable (Flandrin et al., 2004). It therefore provides the basis for a noise elimination approach with results often superior to those of wavelet-based techniques (Kopsinis and McLaughlin, 2009). Recently, EMD analysis has been successfully applied to altimeter data to analyze wave-current interactions known to predominate at scales below 100 km (Quilfen et al., 2018; Quilfen and Chapron, 2019). The main steps of this method are described hereinafter. For a full description of the method, please refer to

10   Quilfen and Chapron (2020).

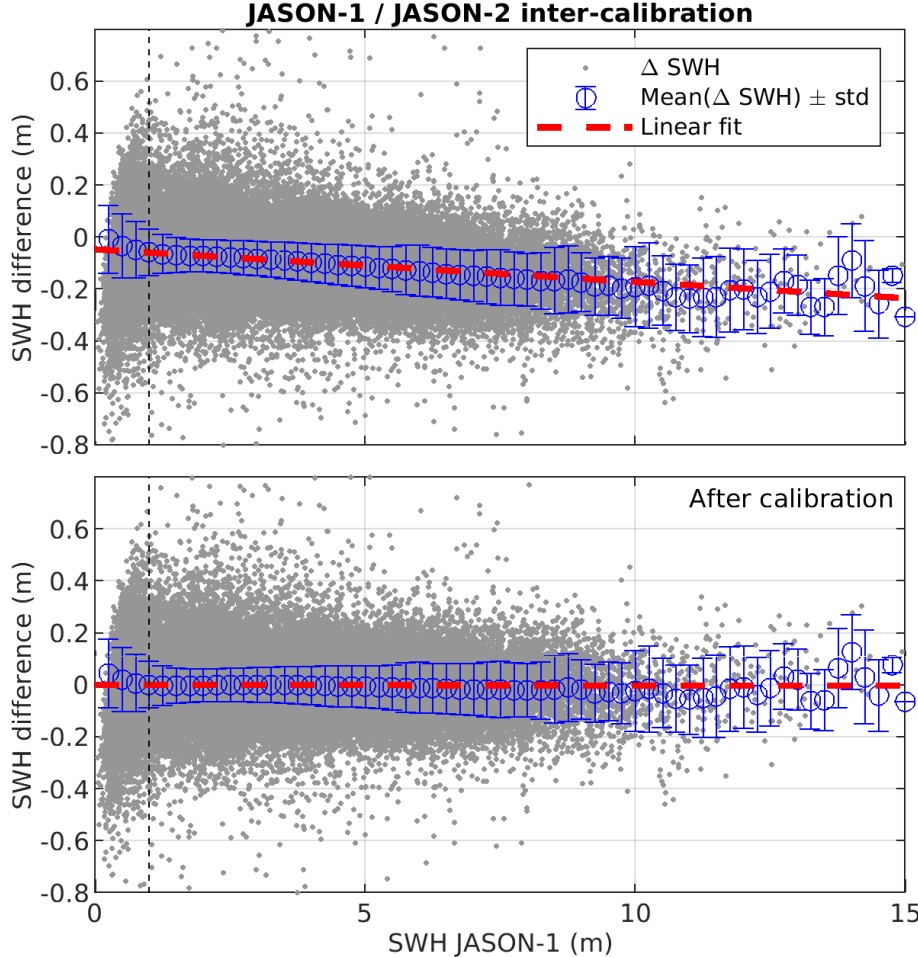

**Figure 4.** Residual of JASON-1 SWH – JASON-2 SWH as a function of JASON-1 SWH before (top panel) and after (bottom panel) calibration. The red dashed line is a linear fit through the data.

### 3.3.1 The EMD principles

EMD adaptively decomposes a signal x(t) into a small number $L$ of IMFs $h_n(t), 1 \leq n \leq L$, so that:

$$x(t) = \sum_{n=1}^{L} h_n(t) \tag{1}$$

The IMF number, $L$, depends on the length of the record and typically varies from 1 to 10 for the lengths analyzed in the altimeter dataset. By construction, IMFs have the following properties: they are zero mean, all their maxima and minima are respectively positive and negative, and they have the same number (or $\pm$ 1) of zero-crossings and local extrema. The IMFs are calculated successively, the first one containing the shortest scales and the last one containing a trend, by construction of the algorithm. Each IMF is estimated using an iterative process called sifting that determines the AM/FM high-frequency part

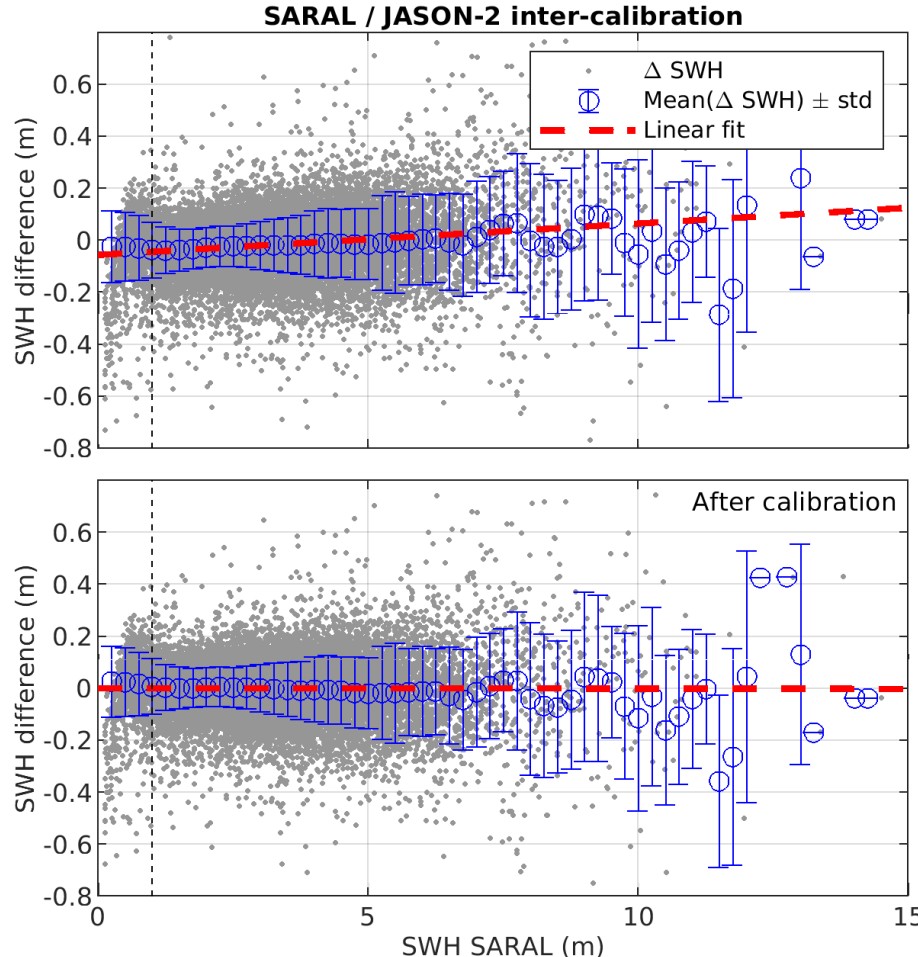

**Figure 5.** Residual of SARAL SWH – JASON-2 SWH as a function of SARAL SWH before (top panel) and after (bottom panel) calibration. The red dashed line is a linear fit through the data.

of any input signal. For a given data segment, the sifting operates in a few steps: 1) find the local maxima and minima; 2) interpolate along the maxima and minima to form an upper and a lower envelope; 3) calculate the average of the two envelopes and subtract it from the analyzed segment; 4) repeat the process from step 1 to 3 unless a stopping criterion has been met (see Huang et al., 1998; Quilfen and Chapron, 2020, for details). An example is shown in Figure 8 for a JASON-2 measurements record of about 1060-km length, for which the EMD method determined six IMFs to represent the full signal. The figure also shows other aspects of the denoising process to be discussed in the next section. As shown, the high-frequency noise is projected in the first IMF, and the scale range of each IMF is increasing with the IMF increasing rank. Notably, the very large geophysical gradients such as observed in this example are also captured by IMF1. IMF1 therefore requires a particular processing to separate noise from useful information.

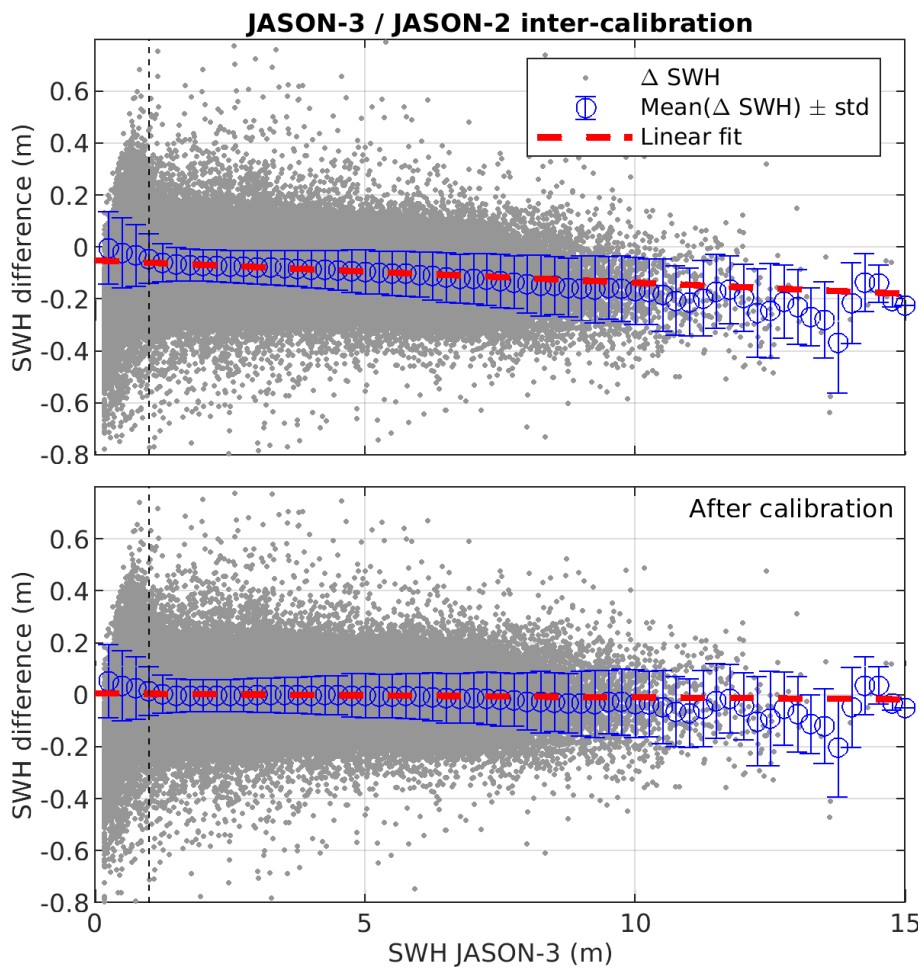

**Figure 6.** (Left) Scatter diagrams of JASON-2 SWH against JASON-3 SWH before (top) and after (bottom) calibration. (Right) Residual of JASON-3 SWH – JASON-2 SWH as a function of JASON-3 SWH before (top) and after (bottom) calibration. The red dashed line is a linear fit through the data.

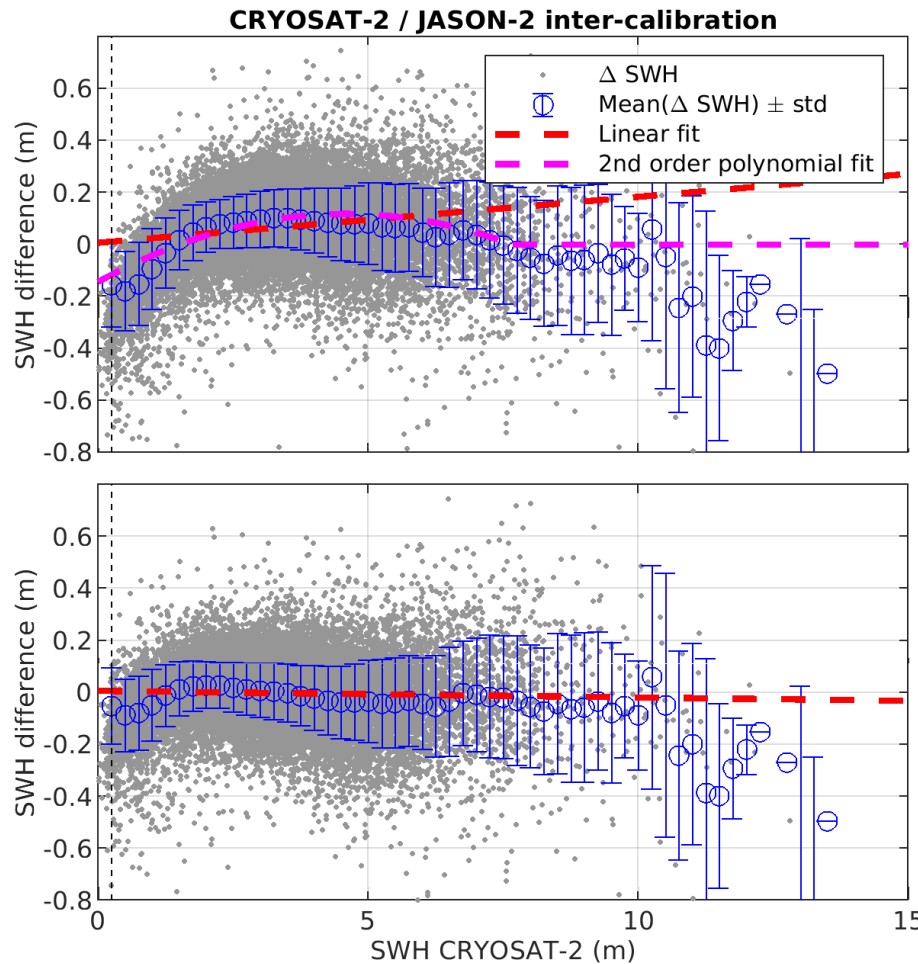

**Figure 7.** Residual of CRYOSAT-2 SWH – JASON-2 SWH as a function of CRYOSAT-2 SWH before (top panel) and after (bottom panel) calibration. The red dashed line is a linear fit through the data.

Once the signal is broken down into a set of IMFs, a denoising strategy inspired by those used for wavelet techniques can be applied. The analysis to be carried out takes advantage of 1) the well-behaved and predictable distribution of Gaussian noise energy with the IMF basis, 2) the legacy of decades of wavelet-based denoising techniques, and 3) an ensemble average approach to estimate a robust noise-free signal.

### 3.3.2 EMD-based data denoising

Flandrin et al. (2004) showed that in the case of pure fractional Gaussian noise, the first IMF possesses the characteristics of a high-pass filter while the higher order modes behave similarly to a dyadic filter bank for which as descending the frequency scale, the successive frequency bands have half the width of their predecessors. This is illustrated in Figure 9.

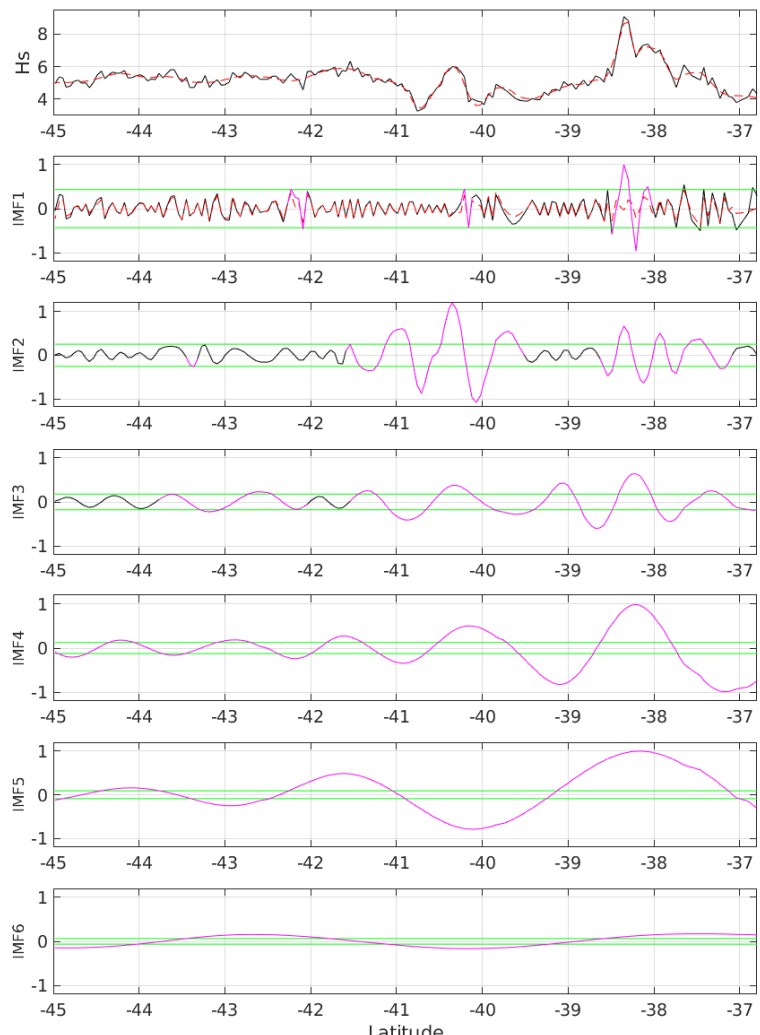

**Figure 8.** EMD expansion in $L$ IMFs (panels 2 to 6 from top to bottom) for a data segment of JASON-2 raw (black solid line) and filtered (dashed red line) SWH measurements on February 29, 2016 (first panel). For the IMFs, the black lines show the IMF amplitudes and the superimposed magenta lines the signal portions identified as above the predicted noise level (green solid lines). For this particular case, portions of signal that are dominated by noise are found solely in the three first IMFs. For IMF1, the dashed red line shows the high-frequency noise series that has been computed from IMF1 wavelet processing. As obtained, six IMFs describe the total signal.

**Table 3.** Calibration formula used for the Sea State CCI dataset v1

| Mission | Calibration formula | Applied to |
|---------|---------------------|------------|
| ERS-1 | $SWH_{cal} = 1.1259SWH + 0.1854$ | All data |
| TOPEX | $SWH_{cal} = 1.0539SWH - 0.0766$ | cycles 0-97 (Side A) |
| | $SWH_{cal} = 1.0539SWH - 0.0766 + dh(cycle)^{(1)}$ | cycles 98-235 (Side A) |
| | $SWH_{cal} = 1.0237SWH - 0.0476$ | cycles > 235 (Side B) |
| ERS-2 | $SWH_{cal} = 1.0541SWH + 0.0391$ | All data |
| GFO | $SWH_{cal} = 1.0625SWH + 0.0754$ | All data |
| JASON-1 | $SWH_{cal} = 1.0125SWH + 0.0461$ | All data |
| ENVISAT | $SWH_{cal} = -0.021SWH^3 + 0.1650SWH^2 + 0.5693SWH + 0.4358$ | $SWH < 3.41\,m$ |
| | $SWH_{cal} = 1.0095SWH + 0.0391$ | $SWH \geq 3.41\,m$ |
| JASON-2 | $SWH_{cal} = 1.0149SWH + 0.0277$ | All data |
| CRYOSAT-2 | $SWH_{cal} = 0.0124SWH^2 + 0.8858SWH + 0.1446$ | $SWH < 7.67\,m$ |
| SARAL | $SWH_{cal} = 0.9881SWH + 0.0555$ | All data |
| JASON-3 | $SWH_{cal} = 1.0086SWH + 0.0503$ | All data |

$^{(1)}dh = -0.0685 + 6.0426.10^{-4}cycle + 7.7894.10^{-6}cycle^2 - 6.9624.10^{-8}cycle^3$

It implies that the Gaussian noise variance projected onto the IMF basis can be modeled, for IMFs of rank $n > 1$, as follows:

$$var(h_n(t)) \propto 2^{(\alpha-1)n} \tag{2}$$

$\alpha$ depends on the autocorrelation function of the fractional Gaussian noise (i.e., $\alpha = 0.5$ for an uncorrelated noise, e.g., white noise; $\alpha \neq 0.5$ for an autocorrelated noise). For white noise, the expected noise energy level of each IMF of rank $n > 1$ is then given by:

$$E_n = \frac{E_1}{0.719}2.01^{-n} \tag{3}$$

where $E_1$ is computed using the Median Absolute Deviation (MAD) from zero:

$$E_1 = \left(\frac{median|n_1(t)|}{0.6745}\right)^2 \tag{4}$$

where $n_1(t)$ is the IMF1 noise estimated from a wavelet analysis (as example see Figure 8, top panel). Eq. 3 and 4 then give the expected noise energy in each IMF to determine the different thresholds below which signal fluctuations are associated with noise, as illustrated in Figure 8. For each IMF, the threshold is $T_n = A\sqrt{E_n}$. $A$ is a constant that can be adjusted as a global tuning parameter.

With the EMD basis, noise energy decreases rapidly with the increasing IMF rank: $\sim 59\%$, $20.5\%$, $10.3\%$, $5.2\%$ of total energy for the first four IMFs, respectively, which represents $\sim 95\%$ of the total noise energy. For a given noisy input signal,

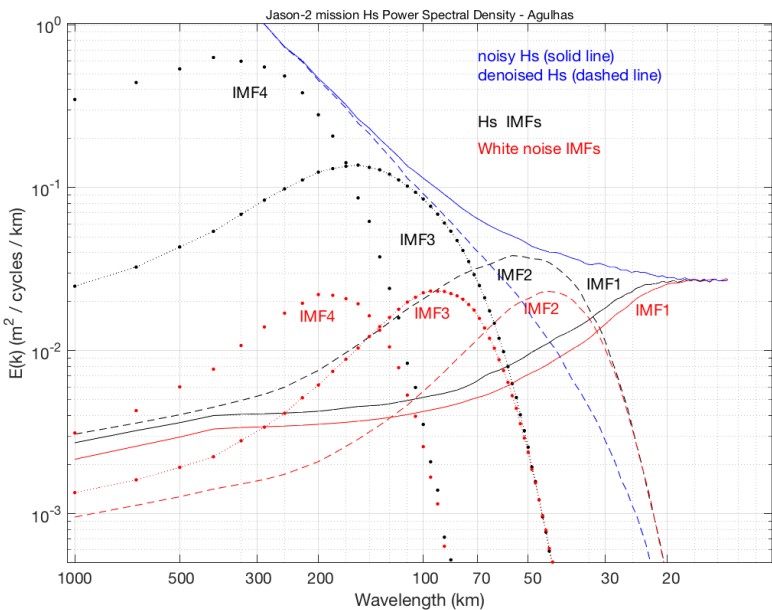

**Figure 9.** Mean Power Spectral Density (PSD) of the first four IMFs for white noise (red curves) and JASON-2 $SWH$ along-track measurements (black curves), and mean PSD of the corresponding noisy (solid blue line) and denoised (dashed blue line) JASON-2 $SWH$ measurements. The PSD is the average of PSDs computed over all data segments covering the years 2014 to 2016 in the Agulhas region (10ºE – 35ºE; 45ºS – 33ºS). From Quilfen and Chapron (2020)

the SNR and robustness of the denoised signal (e.g. to mitigate for result uncertainties associated with signal fluctuations close to the applied thresholds) are increased by estimating the final result as an ensemble average of several denoised signals. For that, the noise $n_1(t)$ is first removed from the noisy signal $x(t)$, then a set of $k$ new noisy signals is generated by adding random realizations of $n_1(t)$, providing after denoising a set of $k$ denoised signals whose average gives the resulting denoised

5   SWH and whose standard deviation gives the uncertainty attached to the denoised SWH. The uncertainty parameter therefore accounts for the noise characteristics of the noisy signal (function of the altimeter sensor, SWH etc) as well as for the local SNR (which is scale-dependent) and for uncertainties attached to the denoising process.

    Figure 9 illustrates the different points discussed above. It shows how the EMD filter bank distributes a white noise signal and the JASON-2 altimeter SWH signal in the Agulhas Current region. The standard deviation of noise was adjusted to fit

10   the SWH background noise at scales < 20 km. As shown, the EMD filter bank is composed of a high-pass filter, IMF1, and a dyadic filter bank for higher ranking IMFs. A similar structure is observed when EMD is applied to the SWH along-track signal, confirming that IMF1 contains mainly the high-frequency noise, and showing that pure noise and SWH higher ranking IMFs share the same frequency ranges. Figure 9, that shows how similar is the filter bank for pure noise and for SWH signal, therefore highlights the practical rule used for denoising, that compares the signal modulation in each IMF with the noise

energy expected for the IMF of same rank. The proposed method is free of systematic artifacts, preserves the amplitude of spatial gradients and extreme values, and eliminates the noise over the whole frequency range. Signals down to scales of nearly 30 km can be recovered, provided that the local signal-to-noise ratio is sufficient.

## 4    Quality assessment of the Sea State CCI dataset V1

5    ### 4.1    Comparisons against in-situ data and model results

Statistical metrics (bias, RMSE, NRMSE, SI and $R^2$) between altimeter measurements and in-situ data were computed for each mission, and each year. The overall scores are provided in Table 4 for the calibrated and denoised altimeter SWH, considering only altimeter-in-situ match-ups that occurred more than 200 km from the coast. With a number of match-up data comprised between 1018 (ERS-1, 3 years of data) and 14395 (JASON-2, 11 years of data), all the computed values are statistically 10    significant. Except for ERS-1 for which the bias is negative (-7.2 cm), all the mission show a positive bias lower than 10 cm. The RMSE is below 26 cm for all missions, corresponding to a mean value lower than 11% once normalized by the mean of the observations. Moreover the scatter index is lower than 9% and the coefficient of determination higher than 0.96 for all missions.

**Table 4.** Statistical metrics for the validation of denoised SWH in the Sea State CCI dataset v1 against in-situ data located >200km offshore

| Mission | N Years | Match-ups | Bias (m) | RMSE (m) | NRMSE (%) | SI (%) | $R^2$ |
|---------|---------|-----------|----------|----------|-----------|--------|-------|
| ERS-1 | 3 | 1018 | -0.07 | 0.26 | 9.95 | 8.41 | 0.97 |
| TOPEX | 12 | 7797 | 0.01 | 0.24 | 9.74 | 8.39 | 0.97 |
| ERS-2 | 17 | 9207 | 0.01 | 0.24 | 10.41 | 8.96 | 0.97 |
| GFO | 9 | 5221 | 0.03 | 0.26 | 10.91 | 9.46 | 0.96 |
| JASON-1 | 12 | 11094 | 0.01 | 0.22 | 9.58 | 8.31 | 0.97 |
| ENVISAT | 11 | 8286 | 0.04 | 0.23 | 10.05 | 8.58 | 0.97 |
| JASON-2 | 11 | 14395 | 0.07 | 0.21 | 9.67 | 7.86 | 0.98 |
| CRYOSAT-2 | 9 | 7913 | 0.07 | 0.20 | 9.17 | 7.46 | 0.98 |
| SARAL | 6 | 7876 | 0.09 | 0.21 | 10.14 | 7.96 | 0.98 |
| JASON-3 | 3 | 4181 | 0.10 | 0.21 | 9.95 | 7.48 | 0.98 |

In order to assess the impact of the calibration and denoising steps applied on the altimeter measurements (Section 3.1), 15    the above-mentioned metrics were also computed for the raw and calibrated SWH data before denoising was applied. Figure 10 shows the averaged bias and normalized RMSE between in-situ and altimeter measurements for the raw, calibrated and denoised SWH. Here again, only match-ups that occurred more than 200 km from the coast were considered. Comparing the statistics for the raw SWH and calibrated SWH, we see that the calibration step tends to decrease the absolute bias and the NRMSE, except for JASON-1, JASON-2, SARAL and JASON-3. In particular, the GlobWave-calibrated JASON-2 measure-

ments present a positive bias of $\sim 8$ cm. Since these data were used to inter-calibrate the JASON-1, SARAL and JASON-3 measurements included in the present dataset, it is straightforward to attribute the positive bias found in these three missions to the propagation of the error during the inter-calibration step. The increased bias also resulted in larger NRMSE after the calibration of these missions. Although a clear understanding of this increased bias for JASON-2 requires further investigations,

the different in-situ dataset (stations and time period) used for the GlobWave calibration and for the present validation may explain part of the discrepancies. Comparing the statistics for the calibrated and denoised SWH, we see that the denoised data compared slightly better with in-situ measurements than un-denoised data, with NRMSE decreasing by up to 7% and by 3% on average, after denoising is applied. The impact of the denoising step was actually much more significant on the comparisons against model outputs, which take into account the along-track variability (see below).

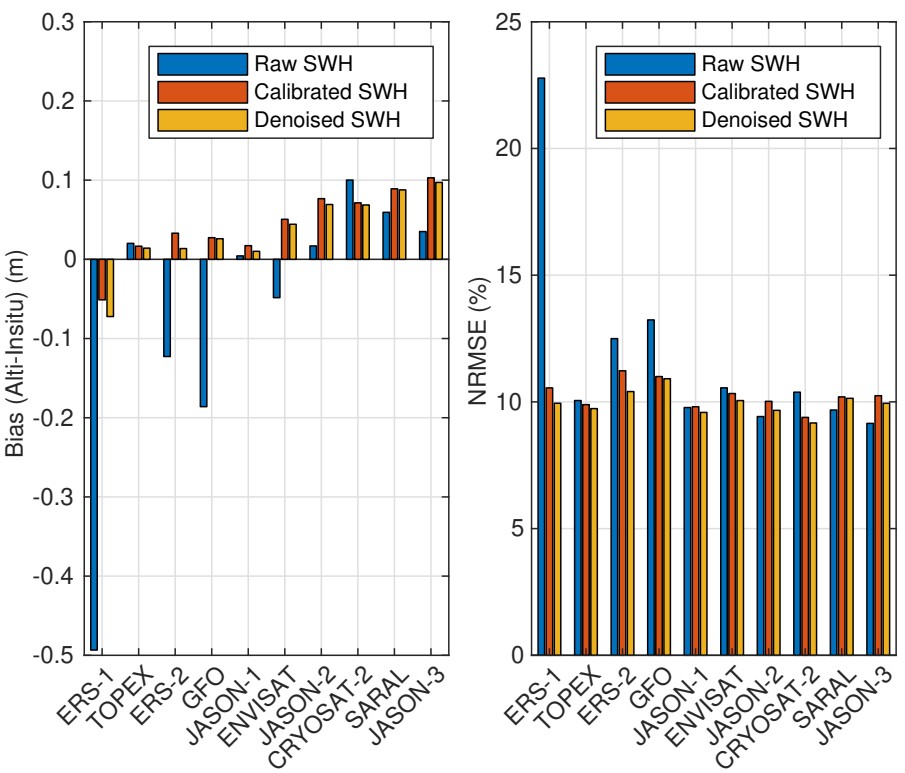

**Figure 10.** Mean bias (left) and mean NRMSE (right) between altimeter measurements and in-situ measurements > 200 km from the coast.

Comparison of the altimeter dataset against the WW3 wave model hindcast (described in Section 2.3) was performed as a complementary validation with an independent dataset. In order to assess the quality of the dataset over the 1994-2018 time period, mean global bias and NRMSE between the denoised altimeter SWH and the modelled SWH were computed on a yearly basis for each altimeter mission. Figure 11 shows the time-series of these two parameters, with a distinct colors for each mission. We can see that the bias is lower than 10 cm and the NRMSE is lower than 13% over the whole period. The overall

trend is a decrease of the error metrics from the oldest missions to the most recent ones that may be attributed to improvements

in instrument performance and processing techniques. We also note some inter- and multi-annual variabilities in the metrics that can be associated with changes in missions recording phases and associated orbits. The thin dashed lines on the bottom panel show the NRMSE obtained before denoising is applied on the altimeter SWH. Differences in the metrics obtained with the calibrated (un-denoised) and denoised SWH illustrate the significant improvements obtained after the small scale (<100km)

5   fluctuations in the altimeter measurements are removed, with a NRMSE decrease by up to 20% and by 10% on average.

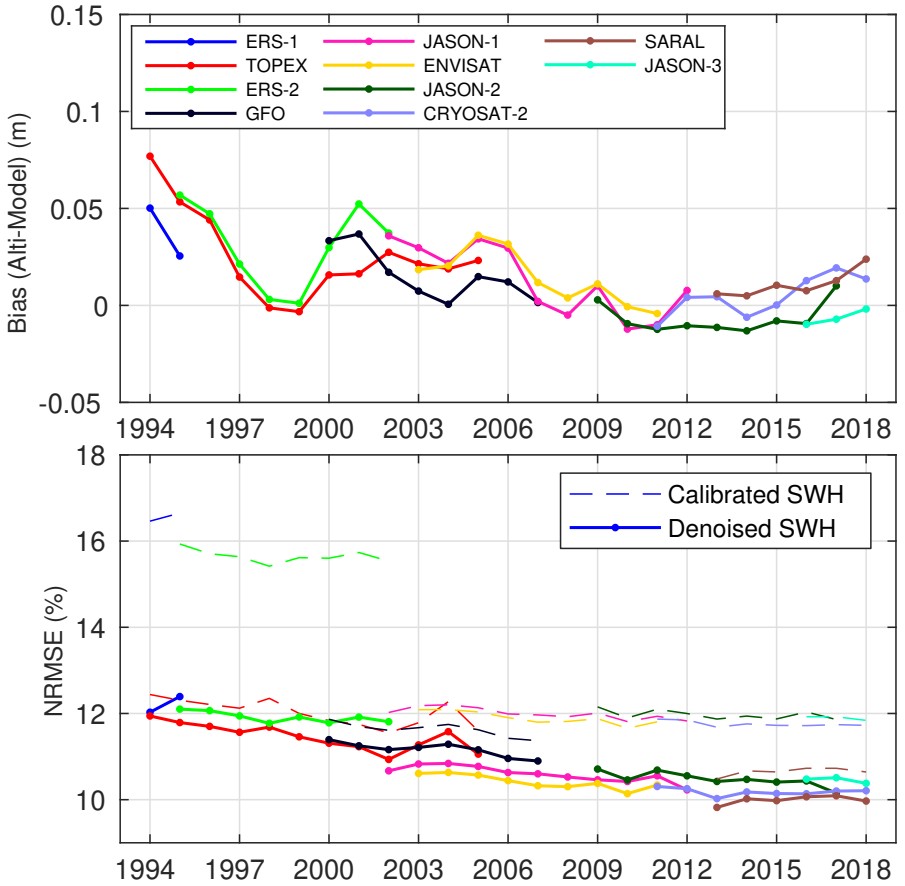

**Figure 11.** Time series of mean global bias (upper panel) and mean global NRMSE (lower panel) between Sea State CCI dataset v1 and WW3 model outputs forced with ERA5 wind fields (see Section 2.3). The thin dashed lines in the bottom panel represent the results obtained for the calibrated SWH before denoising was applied.

Finally, coastal and regional assessments of the dataset were performed by computing error metrics for different coastal strips (>200km, 100-200km, 50-100km, 0-50km) and different basins (North Atlantic, South Atlantic, North Pacific, South Pacific, Indian Ocean, Southern Ocean). The number and locations of the coastal in-situ stations and the (arbitrary) extensions of the ocean basins used for these comparisons are depicted in Figure 1. Due to a low number of in-situ stations in most ocean

10  basins, the regional assessment was only performed against model outputs. Also, given the 0.5° model resolution, coastal

assessment was only performed for model outputs located more than 50 km from the coast. The results of these comparisons are summarized in Table 5. The error metrics for the different coastal strips clearly indicate the better performance for the

**Table 5.** Statistical metrics based on altimeter-buoy and altimeter-model comparisons for different subset of data (see Figure 1).*For the model comparisons, only nodes located further than 50 km from the coast are considered. For the coastal assessment against model outputs, only data within 60°S-60°N are considered.

| Subset | Bias (m) | RMSE | NRMSE (%) | SI (%) | $R^2$ |
|---|---|---|---|---|---|
| Comparisons against buoys | | | | | |
| All | 0.11 | 0.36 | 18.19 | 14.53 | 0.93 |
| > 200 km | 0.04 | 0.23 | 9.96 | 8.29 | 0.97 |
| 100-200 km | 0.07 | 0.30 | 13.42 | 11.02 | 0.96 |
| 50-100 km | 0.06 | 0.28 | 15.81 | 12.76 | 0.95 |
| 0-50 km | 0.16 | 0.44 | 23.69 | 18.27 | 0.89 |
| Comparisons against model* | | | | | |
| All | 0.02 | 0.30 | 11.14 | 10.20 | 0.89 |
| > 200-km | 0.00 | 0.30 | 10.30 | 9.66 | 0.90 |
| 100-200km | 0.08 | 0.29 | 13.22 | 11.45 | 0.87 |
| 50-100km | 0.12 | 0.30 | 16.50 | 13.35 | 0.83 |
| NA | 0.04 | 0.31 | 11.50 | 10.93 | 0.91 |
| SA | -0.02 | 0.31 | 10.64 | 10.11 | 0.89 |
| NP | 0.04 | 0.30 | 11.49 | 10.78 | 0.90 |
| SP | 0.02 | 0.28 | 9.47 | 8.67 | 0.88 |
| IO | 0.06 | 0.23 | 10.67 | 9.62 | 0.90 |
| SO | -0.17 | 0.47 | 13.86 | 11.51 | 0.88 |

comparisons obtained further from the coast. For instance, the NRMSE increases from ∼10% for buoys located >200km from the coast to ∼24% for buoys located less than 50 km from the coast, and from ∼11% for model outputs >200km from the coast to ∼17% for model outputs within 50 and 100 km from the coast. These results also reveal a ∼10 cm increase of the bias between altimeter data and in-situ measurements at less than 50 km from the coast. This increased bias may be explained by the fact that match-ups between altimeter and in-situ stations close to the coast will contain a larger fraction of altimeter records located offshore with respect to the buoy position, the altimeter records nearer to the coast being rejected during the data editing process (see Section 3.1). As a result, the altimeter will systematically see larger waves than the in-situ sensor in the regions where sea states are impacted by coastal features, such as shallow depths, island blocking, or increased tidal currents. For what regard the basin comparisons, we note similar performance for each basins except for the Southern Ocean, where the bias is negative and ∼20 cm lower than for the other regions, and the NRMSE is ∼14% while it is closer to 11% for

the other regions. This may be explained by the poorer quality of both altimeter records and model simulations in this region dominated by high sea states.

## 4.2 Cross-consistency analysis

One objective of the Sea State CCI project is to implement a processing system able to produce accurate and consistent long-term time-series of EO-based SWH measurements. Indeed, the time consistency of the produced dataset is particularly relevant for investigating the multi-decadal variability of the sea states Essential Climate Variable and its interactions with other components of the Earth climate system. In order to ensure that the produced altimeter SWH are consistent over the altimeter time period, we inspected the monthly global mean SWH for each mission, within 60°S and 60°N. Figure 12 shows the time-series of the global monthly means for the raw, calibrated and denoised SWH, with the corresponding mean values computed over the available measurement period. We see that the global means of raw SWH present large differences from one mission to the other, with overall standard deviation ($\sigma$) of the mean values equal to 15 cm. The lowest mean value (2.00 m) is obtained for the earliest mission ERS-1, while the highest mean value (2.54 m) is obtained for CRYOSAT-2. These differences are strongly reduced after calibration is applied to SWH ($\sigma$ = 2.5 cm). We note that denoising the calibrated SWH has a minor impact on the mean values, with maximum changes of the global mean lower than 2%. The remaining differences in the mean values between each mission can be partly attributed to the different calibration methodology applied to the GlobWave dataset and to the most recent missions included in the Sea State CCI dataset v1, but also to the natural variability of the sea states that is partly controlled by inter-annual and decadal climate modes such as the North-Atlantic Oscillation, the El Nino Southern Oscillation, or the Southern Annular Mode (Dodet et al., 2010; Reguero et al., 2019).

## 5 Applications of the Sea State CCI dataset

## 5.1 Global wave height climatology

A first evaluation of the Sea State CCI dataset v1 consists in Figure 13 (top panel) of the global distribution of the climatological annual mean significant wave height calculated over the period 1992-2017. This is based on the CCI Sea State Level 4 (L4) gridded product and is presented here at its native 1° resolution. Further evaluations and analyses of the L4 product over climatological time scales, including intercomparisons with other high quality sea state data sources, are provided in Timmermans et al. (2020). The climatology clearly shows the typical features of global wave fields, with high sea states at mid-to-high latitudes in both hemispheres corresponding to the imprint of extra-tropical storm tracks and the persistently high winds of the Southern Ocean. The 1° resolution of the product makes it possible also to distinguish regions of lower mean wave heights in enclosed and sheltered seas and close to islands and land, for example in the Gulf of Mexico, the Mediterranean Sea, the Baltic Sea and the Indo-Pacific Warm Pool.

Focusing now on the middle panel in Figure 13, we see the normalised climatological difference (expressed as a percentage of SWH) between the CCI product and the climatological mean obtained from the calibrated multi-mission altimeter data

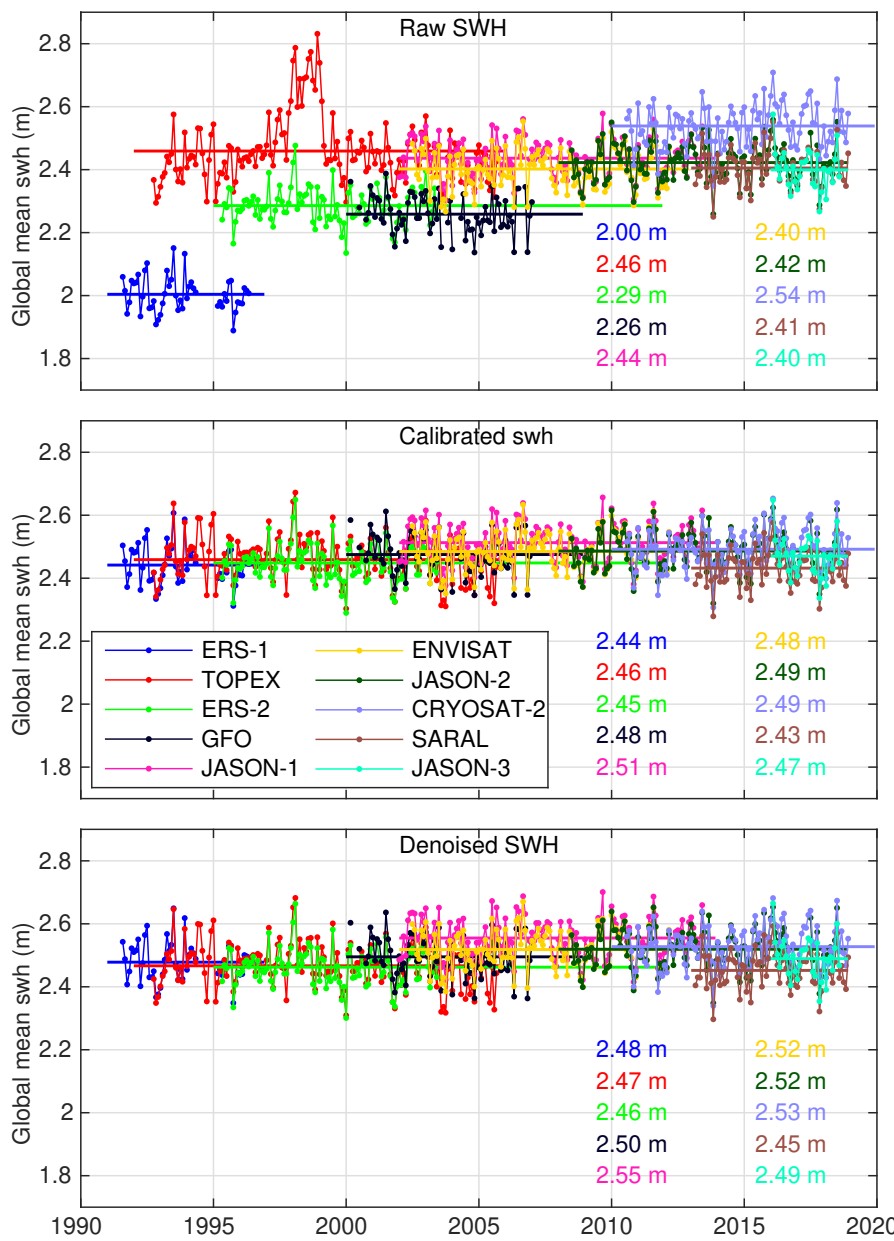

**Figure 12.** Monthly global mean SWH over the period 1991-2018 before (top panel) and after (bottom panel) calibration is applied.

published by Ribal and Young (2019). The overall agreement between the two altimeter-based datasets is generally good, with differences typically less than $\pm 2.5\%$, although some spatially coherent differences (both positive and negative) are clearly visible, most noticeably on either side of the Equator. While a detailed explanation for the differences is subject to further analysis, a number of factors are likely to be relevant. We note some differences in source missions (omission here of Saral,
in Ribal & Young analysis, for example), and differences in the calibration methodology such as the use of different sets of reference data buoys between the two products, and subsequent mission cross-calibration. See Timmermans et al. (2020), section 4 for more details.

Finally, Figure 13 bottom panel presents a similar comparison, this time between the CCI and ERA5. ERA5 (Hersbach et al., 2018) is the most recent of the reanalysis products developed and distributed by ECMWF, that features a number of
innovations, including higher spatial and temporal resolution and hourly assimilation of altimeter significant wave height data. In these results, the comparisons indicate that, even though ERA5 assimilates altimeter data, the ERA5 climatological mean SWH is substantially lower than CCI almost everywhere, except the eastern tropical Pacific and south tropical Atlantic where ERA5 clearly overestimates the wave climate. Once again, as for the comparison against Ribal & Young (2019), strong signatures are observed either side of the Equator. These are likely attributable to at least two factors. Firstly,
ERA5 generally under-estimates SWH in stormy areas, except in the deep tropics where the wave climate is dominated by long period swell. Recent changes have been made to the ERA5 wave physics package to try to solve some of these issues (https://www.ecmwf.int/en/about/media-centre/news/2019/forecasting-system-upgrade-set-improve-global-weather-forecasts). Secondly, in the tropical Pacific Ocean, the impact of the equatorial and counter-equatorial currents, are clearly visible. This corresponds to the absence of ocean surface currents, both in the atmosphere boundary layer and in the wave model component
of ERA5. It is also affected by the relatively coarse (32 km) wind fields, which lead to loss of information in the wave model.

Similar examination of differences in climatological seasonal (JFM, JJA) mean, between the CCI and the product of Ribal & Young, is provided by Timmermans et al. (2020). Differences were generally found not to be statistically significant at a 10% level (their figure S2 in supporting information), with some possible exceptions in regions of low average sea state, such as the Bay of Bengal and Indonesian seas. However, a more rigorous assessment of robustness of differences was recommended,
noting high sea state variability over the relatively short record, and other possible sources of systematic error that remain poorly understood.

## 5.2    Long-term wave height trends

The accurate representation of long term temporal variation is crucial for many applications. Timmermans et al. (2020) examined long-term global seasonal (JFM, JJA) SWH trends from the CCI Level 4 dataset, and intercompared those with other high quality sea state records over the period of continuous satellite coverage (their Figures 3 and S4). Figure 14 shows the trend
in annual mean SWH for the CCI L4 product, with a similar intercomparison. Trends are calculated using a linear regression approach, discussed further by Timmermans et al. (2020).

Overall, there is remarkable variability across datasets, although in all cases intra-dataset variability shows a high degree of spatial coherence. While the maximum range of trends across all datasets is approximately the same, a striking result is

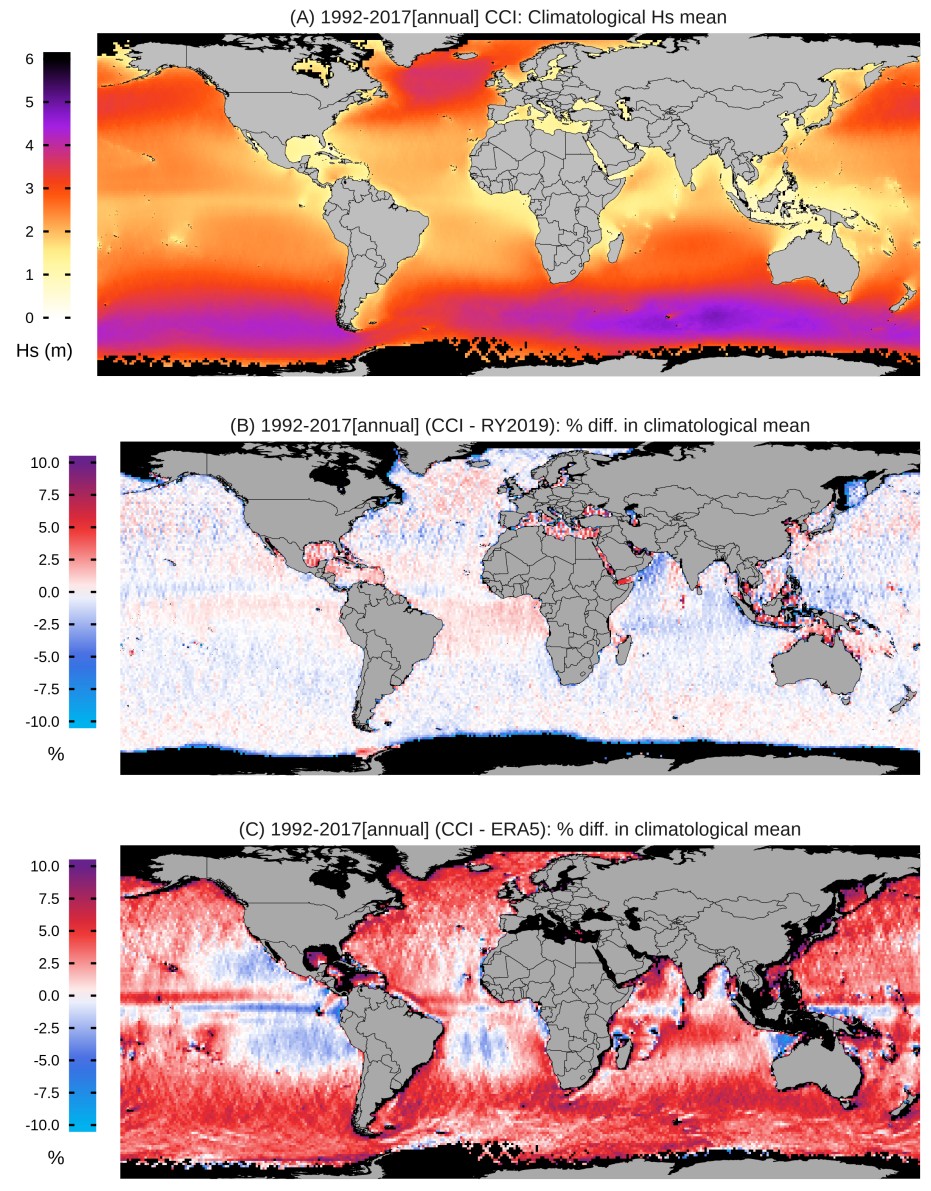

**Figure 13.** (top) Climatological annual mean SWH over the period 1992-2017 obtained with CCI v1 Level 4 data. (middle) Normalised difference (% SWH) between climatological mean from CCI and Ribal & Young, 2019. (bottom) Normalised difference (% SWH) between climatological mean from CCI and ERA5

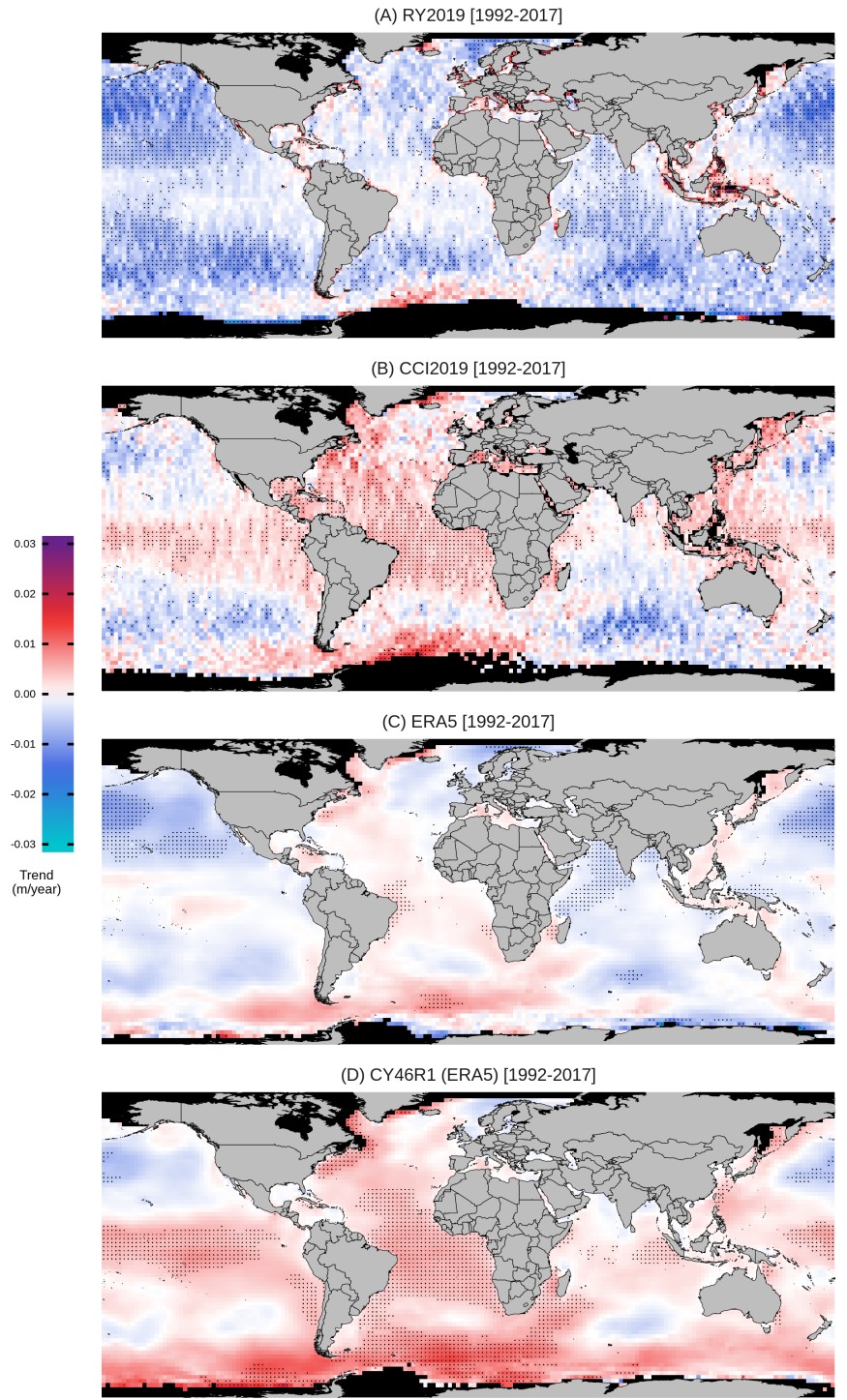

**Figure 14.** Global distribution of annual mean SWH trend estimates on a $2° \times 2°$ grid over 1992-2017 for (A) Ribal & Young (2019), (B) CCI L4, (C) ERA5 and (D) CY46R1 (see Timmermans et al. (2020) for details). Dots indicate grid cells where the trend coefficient is significant at the 5% level.

that trends from the CCI L4 (panel B) appear to be substantially more positive than those from the Ribal & Young (panel A), and in better agreement with the CY46R1 ERA5 hindcast. Some regional intra-dataset trends appear to be robust at the 5% significance level (w.r.t. the linear model) but these are rarely consistent across products, with disagreement in sign in a few locations. CCI L4 contrasts with Ribal & Young with positive trends in the central Atlantic and Eastern Pacific, although there

is qualitative agreement on (negative) sign in the North and South Pacific.

As already highlighted (see also Timmermans et al. (2020) section 4), differences in source missions, and calibration approaches are likely relevant. In particular, Timmermans et al. (2020) reveal (their figure 2) that the CCI dataset tends to provide the largest values (or various datasets, including buoys) in SWH time series at two specific locations, a phenomenon likely linked to the use of Jason-2 as a reference calibration mission. Further factors include the impact of interannual variability

and changes in observation space-time sampling density over the period, both of which may affect the evaluation of a linear trend, particularly if bias is present at the beginning or end of the record. Finally, the relatively high degree of spatial heterogeneity seen for all products suggests that relatively short-term variability has localized influence. In general, on much longer timescales more homogeneous trends might be expected. See also Timmermans et al. (2020) for analysis of seasonal trends (JFM, JJA).

## 5.3  Spectral variability at regional scales

The Sea State CCI dataset v1 provides a unique opportunity to analyze global and regional sea state variability in the mesoscale range below several hundreds kilometers up to ∼50 km. Indeed, the wave field in this scale range is strongly modulated by wave–current interactions (Ardhuin et al., 2017; Quilfen et al., 2018), hitherto neglected in the analysis of altimeter signals due to noise contamination. Moreover, in most ocean basins, altimeter data are the only available measurements of wave

heights. Top panel in Figure 15 shows the yearly averaged mean surface current vorticity computed from altimeter-derived geostrophic surface currents (Rio et al., 2014). Six 1°x1°-regions well exposed to swell events and characterized by different surface dynamics are displayed as colored rectangles. Regions (a) (Agulhas Current, in red) and (b) (Drake Passage, in green) are characterized by strong surface vorticity($> 1x10^{-5}$ s$^{-1}$), with many surface meso- and submesoscale features produced by instability processes (Tedesco et al., 2019; Rocha et al., 2016). Regions (c) and (d) (Atlantic/Pacific equatorial band, in

orange and purple respectively) are characterized by intermediate surface vorticity (between $0.6x10^{-5}$ and $0.4x10^{-5}$ s$^{-1}$). Regions (e,f) (northern branch of South-Atlantic/Pacific gyre, in blue and turquoise respectively) are characterized by low surface vorticity ($<0.2x10^{-5}$ s$^{-1}$).

For each of these regions, we performed a spectral analysis on 8 years (2010-2018) of 1 Hz along-track SWH measurements acquired by JASON-2. The wavenumber spectra were computed along segments of 128 points (∼800 km) detrended and

tapered with a Hanning window. For each region, approximately  3000 1D-spectra were averaged and the mean spectra are shown on the three bottom panels of Figure 15 for each region with similar surface current vorticity (from high vorticity on the left hand side to low vorticity in the right hand side). The divergence between the dotted line (original signal) and the solid lines (denoised signal) highlights the scales at which the SWH variability is dominated by noise. In our case, we used the EMD-based filtering method described in Section 3.3 to reveal the SWH variability at smaller scales. Interestingly, we note

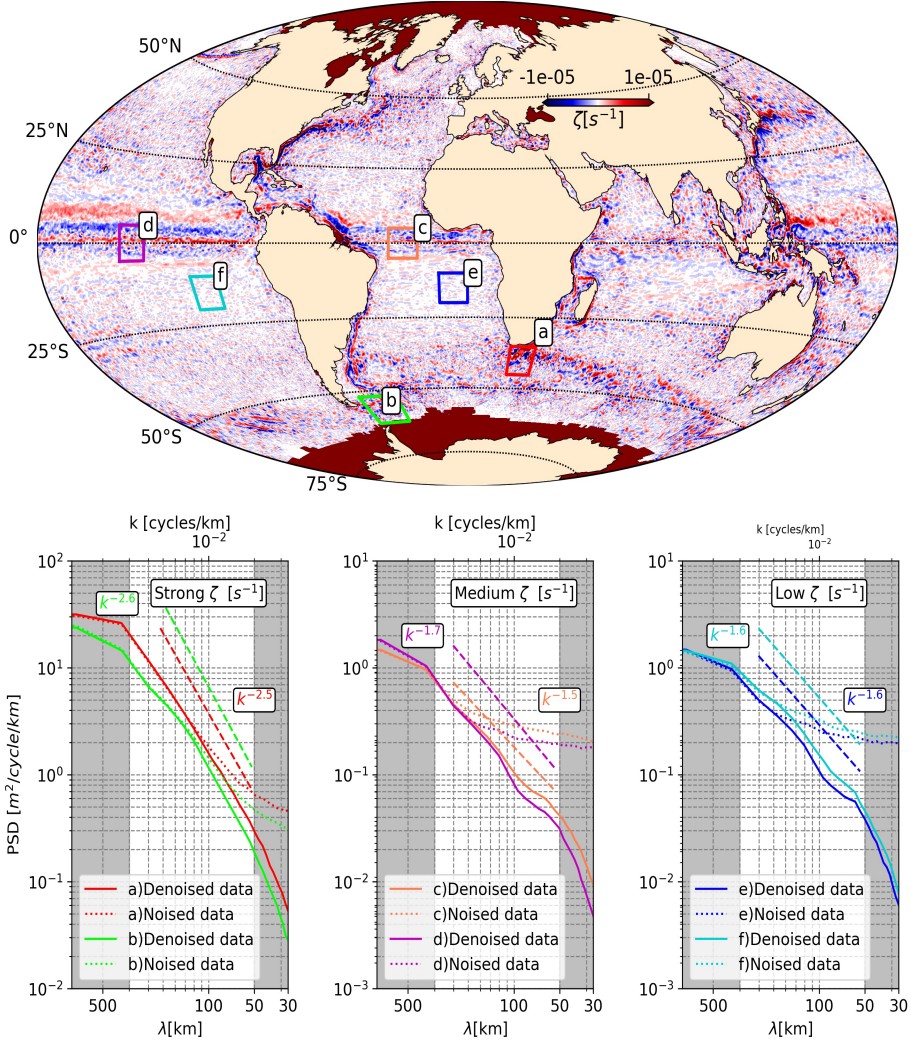

**Figure 15.** (Top panel) Global map of surface current averaged vorticity ($0.25° \times 0.25°$) over 2015 from altimeter-derived geostrophic surface currents (Rio et al., 2014). Colored rectangles are the studied areas through a spectral analysis. (Bottom panels) The associated eight years averaged significant wave height power spectral density (PSD) as a function of wavelength/wavenumber. In solid lines SWH spectra obtained from denoised SWH data and in dotted lines spectra for raw data (including estimation noise). Note that, for reading convenience, the y-axis is not the same for the three bottom subplots.

that the scale at which the divergence takes place depends on the dynamics of the region considered (from ∼125 km for the high vorticity regions to ∼200 km for the lower vorticity regions). Spectral slopes were computed with linear regression over the range 250-50 km (un-shaded area in Figure 15), for which the spectral slope is nearly constant. These slopes are around $k^{-2.5}$ in regions where surface vorticity is intense (i.e Agulhas Current and Drake Passage), as already evidenced by Ardhuin et al. (2017) from combined altimeter data and numerical model results in the Gulf-Stream and in the Drake Passage. In regions where the vorticity is lower, such as the Equatorial band or in the northern branch of Atlantic and Pacific tropical gyres, the spectral slopes becomes less steep (around $k^{-1.5}$). This regional distribution of SWH spectral shape present some similarities with the one obtained for the sea surface height, with steeper slope in high energy area and milder slopes in low energy regions (e.g. Vergara et al., 2019; Xu and Fu, 2011). The difference in the wave height power spectral density between the bottom left panel and the two others is due to the contrasting wave height climatology in the considered regions (see Section 5.1).

These preliminary results highlight the benefit of the EMD-denoising method in order to investigate the small mesoscale SWH variability. Further investigation will be carried out to understand the impact of surface currents on sea state mesoscale variability over multi-decadal scales.

## 6 Current limitations and future developments

This section discusses the current status of the Sea State CCI dataset v1, the main limitations of the data and the perspectives for the future releases of the dataset.

### 6.1 Definition of a reference in-situ dataset for sea states

Routine observations from moored buoys now exceed 40 years for several locations world-wide, which make them practical for analyzing long-term trends. The most abundant open-source network is NOAA's National Data Buoy Center (NDBC) which has maintained an expansive network since the 1970's. Despite the multi-decadal time series from moored buoys, the data homogeneity is a critical issue (Gemmrich et al., 2011). Buoy hulls, payloads, and data processing algorithms change over time and often the changes through meta data are not well documented. The changes in buoy configurations introduce spurious deviations in the time series at least on the same order of magnitude (if not larger) than changes due to inter-annual variability (ENSO, NAO, SAM, etc.) or secular trends. Having detailed metadata is critical to correct the buoy time series. As a result, reported trends from buoy records produce inconsistent results, with changes in magnitude and even sign between buoys separated by a few hundred kilometres (Allan and Komar, 2000; Gower, 2002; Ruggiero et al., 2010; Young et al., 2011). These inconsistencies mean that, at present, very few long-term buoy dataset exist which can be used reliably for trend estimation. This is a major shortcoming as no agreed "ground truth" exists to compare satellite or model estimates of trend. There is a pressing need to produce long-term buoy datasets that include both the measured quantities of interest (significant wave height, wind speed) but also metadata documenting information such as: buoy hull type, sampling details, instrument package, processing details etc. With such metadata, it is potentially possible to correct for changes to such quantities over

time and hence produce a harmonized dataset, in a similar manner to the careful reprocessing of sea surface temperature records (Merchant et al., 2019).

## 6.2 Revisiting altimeter inter-calibration

In its current version (v1), the Sea State dataset uses the GlobWave calibration formula for all missions prior to 2018 (except for JASON-1, as explained in Section 2.1), while the most recent missions were inter-calibrated against the JASON-2 data, as corrected in GlobWave. This strategy was adopted in order to ensure the long-term consistency of this merged dataset. However, the independent validation exercise performed within the CCI project against an exhaustive in-situ dataset (see Section 4) has revealed some unexpected features of the corrected data. In particular, the calibrated JASON-2 SWH presents a positive bias of ∼8 cm, larger than the one computed from the raw data. This discrepancy could be due to the different in-situ data (time coverage and selected networks) used for the GlobWave calibration and the CCI validation. Since JASON-2 is used as a reference for inter-calibrating other missions, this bias also impacts the most recent missions. Moreover, comparisons between the CCI dataset and the one of Ribal and Young (2019) have revealed significant differences in the long-term statistics between these two datasets, which may be partly related to the calibration methodologies and reference in-situ data (Timmermans et al., 2020). Future developments in the CCI dataset will therefore require an improved inter-calibration methodology that will be systematically applied to all altimeter missions included in the dataset, in order to reduce the uncertainties in the long-term sea state statistics.

## 6.3 Assessment and implementation of new retracking algorithms

In order to accurately estimate physical variables of relevance in satellite altimetry, average waveforms (usually at a rate of 20 Hz) are fitted to a mathematical model and an optimization algorithm, in a process called "Retracking". For conventional (low resolution mode) altimetry, all the information on SWH is encrypted in the few bins on the leading edge of the waveform (Figure 16a). The actual echo observed at any bin is the sum of the contributions from many incoherent reflecting points on the sea surface; the effect of this "fading noise" is that the power recorded in the mean waveform has an intrinsic variability that will have a strong effect on parameters calculated from only a few waveform bins. Also, in the coastal zone, unwanted reflections from nearby land or sheltered bays (Gomez-Enri et al., 2010) and changes in the wave shape due to wave-bottom and wave-current interactions (Ardhuin et al., 2012) can affect the quantity and quality of SWH estimations within 20 km of the coast (Passaro et al., 2015). The uncertainty in estimates due to fading noise typically increases with SWH, but the uncertainty is much more pronounced in the near-shore region (Figure 16b). Similar challenges exist in the marginal ice zone. The advent of delay-Doppler altimetry (DDA) offers the potential for improved SWH accuracy near land (Nencioli and Quartly, 2019) and reduced sensitivity to fading noise through being able to utilise a greater number of independent echoes (Raney, 1998).

There is now a strong demand to improve the quality of altimetric wave height data from both LRM and DDA instruments through improved retracking methods in order to: 1) enhance the precision (i.e. short scale repeatability of 20 Hz estimates); 2) increase robustness and accuracy in the coastal zone and ice-affected areas; 3) observe the true spectra of waves unencumbered by retracker resolving issues such as the "spectral hump" (Dibarboure et al., 2014); 4) record accurately the extreme waves

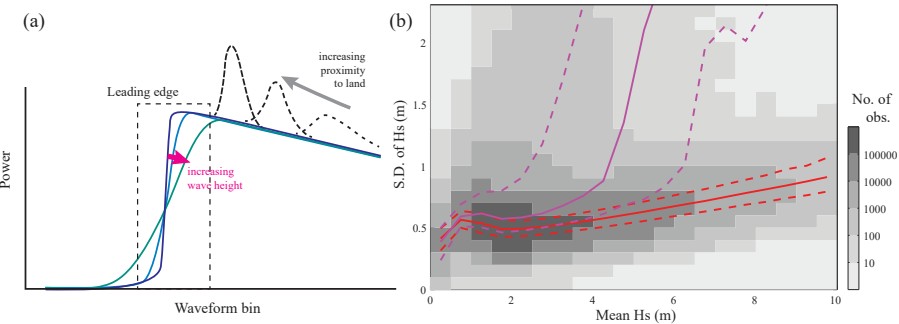

**Figure 16.** (a) Schematic of how conventional waveforms are affected by varying wave height and proximity to land. (b) Variability within a 1-second ensemble as a function of mean conditions. Illustration is from default retracker for JASON-3, with grey shading indicating the population density for open ocean conditions with the red lines indicating the 25th, 50th and 75th percentiles for 0.5 m wide bins. Pink lines show the same analysis for points within 15 km of the coast.

despite uncertainty increasing (Figure 16b); and 5) improve estimation at low SWH where the slope of the leading edge is inadequately resolved (Smith and Scharroo, 2015). DDA shows promise for these aspects although it is worth noting that the much narrower footprint with DDA may be leading to an underestimation bias associated with wave direction (Moreau et al., 2018).

To address these limitations, new retracking techniques have been developed, which generally involve one or more of the following features: numerical solution of the radar equation (as opposed to using an analytical model), fitting of a selected portion of the waveform (Passaro et al., 2014; Thibaut et al., 2017; Peng and Deng, 2018), simultaneous multi-waveform processing (Roscher et al., 2017), and post-processing aimed at reducing correlated errors among consecutive estimations (Quilfen and Chapron, 2020; Quartly et al., 2019; Quartly, 2019). On top of this, several flavours exist of an analytical model

to describe the viewing geometry of the DDA acquisitions (Moreau et al., 2018; Buchhaupt et al., 2018; Ray et al., 2015).

In the framework of the Sea State CCI, a set of rules and statistics for a so-called Round Robin exercise have been defined, which is common in such projects (e.g. Brewin et al., 2015), but to date has never been applied to altimetry. The aim is to ensure that these new algorithms can be evaluated in a rigorous and transparent way, taking into account all the different applications. The procedure involves comparison with external datasets (buoys and models), internal analysis of outlier rejection, quality

flags, precision and spectral properties. The results of this study show that a number of specially designed algorithms can deliver improved SWH retrieval in both open ocean and close to the coast, and for a range of sea state conditions (Schlembach et al., 2020). The gains are achieved both through design of the retracking algorithms e.g. to avoid spurious signals in the tail of the waveform, and also through enhanced data selection using a data quality flag tuned to that specific retracker.

## 7  Conclusions

The Climate Change Initiative program launched by ESA in 2010 has fostered the production of climate-quality long-term global datasets of Essential Climate Variables, whose analysis is needed for understanding the mechanisms of climate change and associated societal impact. In this context, the Sea State CCI project is in charge of reprocessing and developing dedicated algorithms for historical and current EO missions dedicated to the observations of sea state (radar altimeters and SAR missions) in order to produce a continuous, consistent and robust long-term dataset of sea state parameters. The first version of the Sea State CCI dataset, presented in this study, covers the period 1991-2018 and includes observations from 10 altimeter missions. The implementation of quality flags and auxiliary parameters in a systematic way, the update of calibration formula for the most recent missions, the development of an EMD-based denoising method and the validation against an extensive network of in-situ data buoys as well as state-of-the art model results, resulted in a unique dataset designed for the study of wave climate variability. This dataset has already proved really useful to investigate sea state variability at global and regional scales, in terms of wave climatology and spectral variability. Future releases of the Sea State CCI dataset will extend even further the capacity of this dataset, through 1) the implementation of dedicated retracking algorithms for estimating the SWH with improved accuracy; 2) the revision of calibration formula based on a high-quality and consistent data set of in-situ buoys measurements; and 3) the inclusion of spectral wave parameters derived from SAR missions.

## 8  Data availability

The Sea State CCI dataset v1 is freely available on the ESA CCI website (http://cci.esa.int/data) at ftp://anon-ftp.ceda.ac.uk/neodc/esacci/sea_state/data/v1.1_release/. Three products are available: a multi-mission along-track L2P product (http://dx.doi.org/10.5285/f91cd3ee7b6243d5b7d41b9beaf397e1, Piollé et al., 2020a), a daily merged multi mission along-track L3 product (http://dx.doi.org/10.5285/3ef6a5a66e9947d39b356251909dc12b, Piollé et al., 2020b) and a multi-mission monthly gridded L4 product (http://dx.doi.org/10.5285/47140d618dcc40309e1edbca7e773478, Piollé et al., 2020c). The ice masks from SSMI radiometers were obtained from the Centre de Recherche et d'Exploitation Satellitaire (CERSAT), at IFREMER, Plouzané (France).

*Competing interests.*  The authors declare that they have no competing interests.

*Acknowledgements.*  All authors are supported by the European Space Agency under the Sea State Climate Change Initiative project.

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
