# Peer review of "The Sea State CCI dataset v1: towards a Sea State Climate Data Record based on satellite observations"

_Earth System Science Data, 2019_

## Referee Comment (RC1) · Anonymous Referee #1 · 3 Apr 2020

***General comments***

This data set is the output of the "Sea State" project within the Climate Change Initiative (CCI) of the European Space Agency. The paper describes the implementation of the first release of the Sea_State_CCI dataset.

The potential of a consistent long-term data set of sea state data on a global basis is un-questionable. "Sea state" is listed as an Essential Climate Variable (ECV), and is relevant to a wide variety of users, from science to engineering applications.

This project offers three product levels (L2P, L3 and L4) as deduced from satellite radar altimetry, spanning from year 1991 to 2018. L2P is intended as an expert product

containing flagged but un-edited data; L3 (along-track) and L4 (gridded) are higher level products, obtained after systematic calibration and merging between the different satellite altimetry missions used, taking Jason-2 as reference.

The data set builds on the experience of the previous Globwave project (which essentially offers an L2 product), thus carrying a mature methodological background. The accompanying documentation is adequate in describing the data organisation and methods. No problems in accessing and downloading the data that I picked up, by probing various missions, times, and processing levels.

Differently from Globwave, the Sea_State_CCI products include de-noised SWH data obtained by a non-parametric denoising method (EMD – Empirical Mode decomposition). In addition to multiple missions cross-overs and buoy match-ups, this data set also introduces an interesting idea of validation against numerical model outputs, which is described in the submitted manuscript.

A key aspect in the delivery of products destined to multiple user communities, like in this case, is a clear description of the dataset. A well documented and consistent manuscript, together with an easy accessibility to the data are fundamental when dealing with diversified users, characterised by various degrees of expertise in handling the data. In my opinion, the manuscript satisfies this requirement in general, and just needs few technical edits and some clarification, as mentioned later.

Another important aspect in such user-oriented products is the need for clear and trustable indicators of the quality of the data, which should desirably be as complete as possible. Under this point of view, I think that the calibration and validation compartment of this manuscript still has some room for further expansion.

Said that, my overall impression about this work is very positive. The main action that I recommend regards an expansion of Annex B, concerning validation. Other aspects are very minor and mostly technical.

***Specific comments (scientific)***

The usage of Jason-2 data set as a reference for calibration by satellite crossovers is declared to be "evolving" at page 53 (Annex A). It would be interesting if the authors could specify which missions, data sets or techniques they plan to use next. This may seed a useful discussion with the community, at the benefit of the project.

Also, it would be interesting to know if the authors plan to give access to buoy match-up data. In particular, users may take profit from single-buoy-based match-up files (i.e. all the suitable corrected SWH data from proximal satellite tracks vs a given buoy), and do their own validation exercises.

Connected with the above, authors summarise results of the match-ups for all the missions in Table 1 (Annex B) and show bias and NRMSE plots calculated on a global basis in Figure 8. It would be interesting if the authors could provide statistics on a regional basis, too. In particular, a user may be interested in checking for differences in bias and NRMSE between different regional seas. Same applies to the expected SWH uncertainties as defined in section 4.1.2.4, which seem to be averaged and globally applied in a homogeneous way.

Authors at page 57 state that "…the validation of altimeter SWH was performed on a reduced data set including only offshore buoys", the threshold being set at 200km to the coast. Similarly to the comment above, it may be interesting to offer separate statistics for more "coastal" buoys, where users can take SWH estimations even accepting a degraded accuracy when getting closer to the coast. Again, please consider the possibility to split this analysis regionally wherever it makes sense, e.g. depending on the density and distribution of the available buoys.

I hope that these points may seed useful discussion and contribute to improve this new and relevant dataset.

***Specific comments (dataset organisation and processing)***

[Figure]

Section 4.1.4 lists a series of planned improvements foreseen for the next releases of this dataset. Please clarify if the time series extension beyond 2018 and the incorporation of SRAL data will regard SWH only or Sea Surface Height as well, which is currently limited to Feb 2016. This would be interesting for next releases.

In section 6 the SWH outlier test is described in many sub-sections. I suggest to add some brief explanation for justifying the thresholds (5 std dev and 5m).

Regarding the L3 data set, a table with the assignment of numeric identifiers to each satellite seems missing in the document. I found such information only in the .nc file in the flag_values and flag_meanings fields of the "satellite" variable attributes. I suggest to add it to Section 4.2. I suggest also to specify in the text that the flag_values field contains the identifiers of the satellites named in the corresponding positions of the flag_meanings field. This is not necessarily obvious for the non-expert user.

I enclose pictures generated by projecting data extracted from a Jason-2 track (1 January 2018) onto a kml file. This "case study" is in South America, with the track approaching the coast with a high inclination, and crossing a narrow gulf before entering inland. This was chosen to check point flagging and rejection. I enclose pictures of SWH and SWH_denoised, the latter showing a reasonable rejection of close-to-coast points, especially when the track approaches the coastline in an almost-parallel way. In L2P data, SWH adjusted and its uncertainty are calculated in all points where SWH is available, being SWH directly copied from the original GDR files. This produces estimations also in datapoints labeled as "bad" and even flagged as "not_water", as clearly seen in the pictures. The third plot shows the footprint of all the available L3 data related to the same day. Focusing on the study area, data seem to be correctly edited with only valid measurements retained. I also had the chance to check a crossover in the Pacific Ocean, West of this area, finding a good agreement between the two missions involved. This is not a proof of anything of course, being an overall error analysis already conducted by the authors. It was just intended as a random consistency check.

\*\*\*Technical corrections\*\*\*

Here are a few points on more detailed technical aspects, listed as follows.

The work by Quilfen and Chapron, identified as "2019b" in section 4.1.2.5 is fundamental to describe the denoising technique, but still unpublished at the time of writing the submitted document. Please state if it's currently accepted and if a pre-print is obtainable from the authors.

There are still some typos around, and a further general grammar check would improve the document, such as:

page 8: "...The processing of some missions may relies on older GDR versions"–>"...may rely..."

Page 10: "...improves the GlobWave products which were..." –> "...improves the GlobWave products, which were..."

Page 23: "...adjusted indirectly by a first comparison with ERS-2, itself adjusted relatively..." –> "...adjusted indirectly by a first comparison with ERS-2, which is in turn adjusted relatively..."

Page 53: "This calibrated Jason-2 swh data is considered..." Despite the usage of "data" as singular noun is tolerated today, may I suggest the more common plural usage.

Another few suggestions regarding the text are listed here:

page 18: the variable "swh_rejection_flag" formalised in the meta-description table at page 19, is called "swh_rejection_flags" at page 18. Please remove the inconsistency and check for other similar situations through the text.

Page 23: "...ENVISAT sigma0, which seems to be stable with time..." Please try to add a couple of rows to justify this sentence, which looks too much qualitative.
Page 27: "...or could be matched to the L2P measurements for some reason." It may be interesting here to list briefly the main reasons considered for rejection.

Page 45: the sentence "Besides, a test on swh rms (as provided in GDR for 1 Hz measurements) is performed, checking it is below an altimeter and swh dependant threshold" is actually unclear.Please rephrase and try to be more specific.

[Figure]

**Fig. 1.**

[Figure]

**Fig. 2.**

**Fig. 3.**

---

## Referee Comment (RC2) · Anonymous Referee #2 · 14 Apr 2020

Review of 'The sea state CCI dataset v1: towards a sea state climate data record based on satellite observations' by Guillaume Dodet et al.

The authors describe a new sea-state dataset, for which they cross-calibrated ten satellite radar altimeters. They do a rigorous validation of the resulting dataset against another altimetry-derived dataset, in-situ data and wave models/reanalysis. Then the applications of the dataset are described and finally a list of shortcomings is provided.

Major comments

I think the structure is clear and the manuscript is easy to follow. The results are quite convincing, because of the extensive validation. However, I have the feeling that sections 4 and 5 lack depth. In most cases an observation is made, but the details or a possible explanation are never provided. It therefore gives the manuscript the character of a technical report. Therefore I have to recommend a major revision.

Section 5 holds a discussion of applications, but there is only a comparison with other sea-state datasets and a hind that the geostrophic vorticity is connected to wave spectra. This should be expanded and rewritten.

Section 6.2 discusses the implementations of new retracking algorithms, but does not really provide the potential for near-future improvement. I relies largely on the delay/Doppler, but this only helps CryoSat-2, Sentinel-3 and Sentinel-6, which is only from 2010 onwards. Whatever retracking algorithm you implement, you will only find limited improvement in sea state near the coast.

General comments

Make sure that in the captions clear information is provided. For example: in figure 10 a 'wave model hindcast' is used, but it is not directly clear which model is used here.

Minor comments
Page 2.
Line 1: As this is not a oceanography journal, remind the reader what is meant with sea state.
Line 14: I do not see how an acceleration in sea level would translate into changes in sea state, except maybe for coastal waters. A line of explanation or a reference needed.
Line 30: Remove the dot between cm and year.

Page 5.
Line 4-19: The time series are filtered with a 1-hour filter, but most time series have a sampling rate of an hour of more. Should it be interpolated? Be also aware that in coastal zones an hour is quite long, so the interpolation/filtering method is quite important.

Line 18: I think the validation should be shown for both, because you mention in the introduction that sea state is important for coastal processes.

Page 8.
Line 25: The figures seem quite convincing, but I wonder if there is latitudinal dependence remaining in the SWH differences, or even a water-depth dependence. The CCI sea-surface height products are corrected for a latitudinal dependence, which does not necessarily have to be orbit as we also observe it between Jason-1&2.

Line 30: Are you considering only LRM data from CryoSat-2? If not, how is the SAR mode data processed: as Pseudo LRM or delay/Doppler?

Page 11.
Table 3: I would split TOPEX-A&B, because the instruments show clearly different properties. For TOPEX-A: is the cal1 applied or not and is it consistent with TOPEX-B?

Page 12.
Line 23: "The IMFs … algorithm." I think there is something missing in this sentence, maybe: "If the IMFs .. algorithm."

Page 14.
Line 13: For white noise.

Page 16.
Line 2: Why is there a negative bias for ERS, I would expect that the biases are removed by the cross-calibration? There is also a positive bias for the other missions, is this a problem in the data or the model?

Page 17.
Merge section 4.1 and 4.2, it is very short.

Page 18.
Figure 12: A bias between TOPEX and Jason-1 appears to be introduced in the lower figure.

Figure 12: Maybe I missed it, but I am not sure it is clear how the ERS series is de-biased with respect to the Jason series. Maybe this can be added to table 3.

Line 13-18: A bit more in-depth discussion is required why the differences are there. Is it related to the filtering or the de-biasing for example? The reader should know which dataset they should use when they have a certain application in mind. Also the number of 2.5% looks quite nice, but as both datasets are coming from the same dataset I am not sure this is small.

Page 19.
Line 5: I have the feeling that an issue in ENSO, PDO and IOD modeling are affecting the climatology. The authors have a strong oceanography background in the team. It should be possible to speculate at least what is causing this.

Section 5.1: What I am missing here is a comparison of geographical trends and a comparison of the variability. As mentioned in the introduction, changes in sea state are important for a variety of reasons, but the change is completely left out of the discussion. It will probably also give you more insight into the previous comment.

Line 10: For illustration purposes.

Section 5.2: This section suggests that the wave energy is higher in high-geostropic-vorticity regions. Instead of showing a couple of examples, I would compute an average spectrum over low-, medium- and high-vorticity regions. The authors show a correlation between the two quantities, but do not discuss how both are affecting each other. I am not sure what I should get from this section, it should be further elaborated. The topic also seems a bit selective.

Page 21.

Figure 14: The figure for vorticity is too small.

Line 13: Explain fading noise.

Line 15: Quantity and quality.

Line 13-15: One of the major issues is the change of wave shape in shallow areas, which is not discussed here. This is not modeled in standard retrackers.

Line 22: Multilooking is already applied to reduce speckle in LRM waveforms. Typically 90 waveforms are averaged. The great benefit of delay/Doppler is the enhanced along-track resolution, which allows you to get closer to the coast under certain angle-of-attacks. On top of that it increases the number of independent looks per second by a factor of +/-2, which can be used to improve precision (reduce speckle).

---

## Referee Comment (RC3) · Anonymous Referee #1 · 17 Apr 2020

May I add a brief note about the paper. Actually, I have very few things to add respect to what I already said about the product user guide accompanying the dataset, and about the dataset itself.

It's a clear and well readable paper. Differently from the accompanying document, this paper looks almost immune from typos and grammar errors. My comments are substantially reflecting my impression in analysing the dataset and its accompanying doc, which I revised few days ago.

In my opinion the explanations given in Section 3 are convincing, in particular I like the description of the de-noising method. Also, I find Fig. 9 very informative, but, you'll find

just one question about it below (in "Technical points").

I still think that calibration/validation need further expansion (Section 4). Under this point of view, the science paper can adhere to the expansion suggested for the product user guide.

Also, for what regards Section 5, a further discussion of the regional trends identified in Sect. 5.2 may be helpful to many potential users.

Here I have some technical points

- Page 2, lines 12-15: please provide either two lines of explanation or cite a reference to justify the effect on sea state.

- Page 5, lines 17-18: as I suggested in the previous note, characterization of SWH for buoys < 200km can be interesting fpr many users.

- Page 14: I'm not convinced by the x-axis notation (wavenumbers) in Fig. 9: does it span 1000 to 20 cycles/kmgoing left-to-right? Seems contradicting the bandwidth reduction for higher order modes.

- Page 15, line 23: is the subject of the sentence "It highlights. . ." still "Figure 9", mentioned 5 lines earlier? Please put an explicit subject.

- Pages 17-18: please note that legends and axes' texts are hard to read in Figures 10 and 12.

- Page 18 lines 13-17: If you calculate and show the differences between CCI and the dataset by Ribal and Young, you make the reader curious at least to get some conjecture about reasons to justify the differences. I suggest at least to specify the main differences (e.g. filetring) there.

- Page 19 line 25: ". . .several locations worldwide which" –> ". . .several locations worldwide, which. . ."

- Page 21: Fig. 14 is particularly small, especially the left pane individuating the regions of interest.

- Page 21 line 3: "...dataset exit..." –> "...dataset exists..."

- Page 21 line 5: "...buoy datasets which include..." –> "...buoy datasets that include..."
* * *

---

## Author Comment (AC1) · 17 Jun 2020

**Referees' comments**
Our response
*Change in the revised manuscript*

**Response to Referee Comments 1 and 3 (from Referee #1)**

**RC1**

General comments
This data set is the output of the "Sea State" project within the Climate Change Initiative (CCI) of the European Space Agency. The paper describes the implementation of the first release of the Sea_State_CCI dataset.

The potential of a consistent long-term data set of sea state data on a global basis is un-questionable. "Sea state" is listed as an Essential Climate Variable (ECV), and is relevant to a wide variety of users, from science to engineering applications.

This project offers three product levels (L2P, L3 and L4) as deduced from satellite radar altimetry, spanning from year 1991 to 2018. L2P is intended as an expert product containing flagged but un-edited data; L3 (along-track) and L4 (gridded) are higher level products, obtained after systematic calibration and merging between the different satellite altimetry missions used, taking Jason-2 as reference.

The data set builds on the experience of the previous Globwave project (which essentially offers an L2 product), thus carrying a mature methodological background. The accompanying documentation is adequate in describing the data organisation and methods. No problems in accessing and downloading the data that I picked up, by probing various missions, times, and processing levels.

Differently from Globwave, the Sea_State_CCI products include de-noised SWH data obtained by a non-parametric denoising method (EMD – Empirical Mode decomposition). In addition to multiple missions cross-overs and buoy match-ups, this data set also introduces an interesting idea of validation against numerical model outputs, which is described in the submitted manuscript.

A key aspect in the delivery of products destined to multiple user communities, like in this case, is a clear description of the dataset. A well documented and consistent manuscript, together with an easy accessibility to the data are fundamental when dealing with diversified users, characterised by various degrees of expertise in handling the data. In my opinion, the manuscript satisfies this requirement in general, and just needs few technical edits and some clarification, as mentioned later.

Another important aspect in such user-oriented products is the need for clear and trustable indicators of the quality of the data, which should desirably be as complete as possible. Under this point of view, I think that the calibration and validation compartment of this manuscript still has some room for further expansion.

Said that, my overall impression about this work is very positive. The main action that I recommend regards an expansion of Annex B, concerning validation. Other aspects are very minor and mostly technical.

We thank Reviewer #1 for their constructive comment (Referee Comments 1 and 3). These comments concern either the Sea State CCI Product User Guide (RC1) or the submitted manuscript (RC2), but apply to both documents in many cases. We have therefore merged our responses to RC1 and RC3 in a single document. Note that we did not consider some of the comments that only applied to the Product User Guide. In that case, it is mentioned in our response. As suggested, we have strongly improved Section 4 on the validation of the Sea State CCI dataset. In particular, we have included coastal and regional assessments based on buoy and model comparisons. Following your specific comments and the ones from Referee #2, we have also expanded Section 5 and Section 6, and improved the overall manuscript. Please find below our point-by-point response to your comments. All changes in the manuscript can be found in the track-changes manuscript provided at the end of this document.

**Specific comments (scientific)**

**The usage of Jason-2 data set as a reference for calibration by satellite crossovers is declared to be "evolving" at page 53 (Annex A). It would be interesting if the authors could specify which missions, data sets or techniques they plan to use next. This may seed a useful discussion with the community, at the benefit of the project.**

This assertion is given in the Product User Guide but not in the manuscript. Yet, an additional section has been added in the revised manuscript (Section 6.2) in order to discuss the need to improve the calibration procedure. Since the new methodology is currently under investigation, no specific details have been provided in the manuscript. However, discussion on this topic with the user community is clearly sought for and will be addressed during future User Consultation Meetings of the CCI project.

**Also, it would be interesting to know if the authors plan to give access to buoy match-up data. In particular, users may take profit from single-buoy-based match-up files (i.e. all the suitable corrected SWH data from proximal satellite tracks vs a given buoy), and do their own validation exercises.**

Thank you for this suggestion. In the current version of the dataset, the match-up database is not available, but we will think of the best way to include it in future products.

**Connected with the above, authors summarise results of the match-ups for all the missions in Table 1 (Annex B) and show bias and NRMSE plots calculated on a global basis in Figure 8. It would be interesting if the authors could provide statistics on a regional basis, too. In particular, a user may be interested in checking for differences in bias and NRMSE between different regional seas.**

New validation results have been computed for each of the following basins: North Atlantic, South Atlantic, North Pacific, South Pacific, Indian Ocean, Southern Ocean, in order to provide a regional assessment of the product's performance. These results have been computed through comparisons between altimeter data and model results only as in-situ observations are unequally distributed within each basin. These results are given in Table 5 and discussed in Section 4.

**Same applies to the expected SWH uncertainties as defined in section 4.1.2.4, which seem to be averaged and globally applied in a homogeneous way.**

This comment applies to the Product User Guide and not to the present manuscript. Dedicated studies on the regional assessment of SWH uncertainties will be undertaken during the course of the Sea State CCI project.

**Authors at page 57 state that ". . .the validation of altimeter SWH was performed on a reduced data set including only offshore buoys", the threshold being set at 200km to the coast. Similarly to the comment above, it may be interesting to offer separate statistics for more "coastal" buoys, where users can take SWH estimations even accepting a degraded accuracy when getting closer to the coast. Again, please consider the possibility to split this analysis regionally wherever it makes sense, e.g. depending on the density and distribution of the available buoys.**

Validation results have been computed for various coastal strips (0km-50km, 50km-100km, 100km-200km, >200km), both from in-situ observations and from model results. These results highlight the lower quality of altimeter data near the coast. These new results are summarized in Table 5 and discussed in Section 4.

**I hope that these points may seed useful discussion and contribute to improve this new and relevant dataset.**

We clearly think they do. Thanks about it.

**Specific comments (dataset organisation and processing)**

**Section 4.1.4 lists a series of planned improvements foreseen for the next releases of this dataset. Please clarify if the time series extension beyond 2018 and the incorporation of SRAL data will regard SWH only or Sea Surface Height as well, which is currently limited to Feb 2016. This would be interesting for next releases.**

The time-series extension beyond 2018 and the inclusion of SRAL data will regard SWH only. Sea Surface Height time-series will keep being provided within the Sea Level CCI dataset for which time-series extension may be performed in future releases. Since this comment only concerns the Product User Guide, no particular change in the manuscript has been done.

**In section 6 the SWH outlier test is described in many sub-sections. I suggest to add some brief explanation for justifying the thresholds (5 std dev and 5m).**

These thresholds were established through careful visual inspection of the data. Future improvements of the dataset will include a more systematic definition of these thresholds. The paragraph has been modified as follows:

*measurements that deviate from the 100-km mean (excluding the two most extreme values in the mean calculation) by more than five standard deviations or by more than five meters are discarded. These empirical thresholds were defined through careful visual examination of the data.*

**Regarding the L3 data set, a table with the assignment of numeric identifiers to each satellite seems missing in the document. I found such information only in the .nc file in the flag_values and flag_meanings fields of the "satellite" variable attributes. I suggest to add it to Section 4.2. I suggest also to specify in the text that the flag_values field contains the identifiers of the satellites named in the corresponding**

**positions of the flag_meanings field. This is not necessarily obvious for the non-expert user.**

Thank you for this useful comment. More detailed information on the satellite identifiers will be provided in future release of the Sea State CCI dataset and associated Product User Guide.

**I enclose pictures generated by projecting data extracted from a Jason-2 track (1 January 2018) onto a kml file. This "case study" is in South America, with the track approaching the coast with a high inclination, and crossing a narrow gulf before entering inland. This was chosen to check point flagging and rejection. I enclose pictures of SWH and SWH_denoised, the latter showing a reasonable rejection of close-to-coast points, especially when the track approaches the coastline in an almost-parallel way.**

**In L2P data, SWH adjusted and its uncertainty are calculated in all points where SWH is available, being SWH directly copied from the original GDR files. This produces estimations also in datapoints labeled as "bad" and even flagged as "not_water", as clearly seen in the pictures. The third plot shows the footprint of all the available L3 data related to the same day. Focusing on the study area, data seem to be correctly edited with only valid measurements retained. I also had the chance to check a crossover in the Pacific Ocean, West of this area, finding a good agreement between the two missions involved. This is not a proof of anything of course, being an overall error analysis already conducted by the authors. It was just intended as a random consistency check.**

Thanks for checking!

**Technical corrections**
**Here are a few points on more detailed technical aspects, listed as follows.**
**The work by Quilfen and Chapron, identified as "2019b" in section 4.1.2.5 is fundamental to describe the denoising technique, but still unpublished at the time of writing the submitted document. Please state if it's currently accepted and if a pre-print is obtainable from the authors.**

This article has now been accepted. The correct reference is provided in the revised version of the manuscript.

**There are still some typos around, and a further general grammar check would improve the document, such as:**
**page 8: ". . .The processing of some missions may relies on older GDR versions"–>". . .may rely. . ."**
Done

**Page 10: ". . .improves the GlobWave products which were. . ." –> ". . .improves the GlobWave products, which were. . ."**
Done

**Page 23:** ". . .adjusted indirectly by a first comparison with ERS-2, itself adjusted relatively. . ." –> ". . .adjusted indirectly by a first comparison with ERS-2, which is in turn adjusted relatively. . ."
Done

**Page 53:** "This calibrated Jason-2 swh data is considered. . ." Despite the usage of "data" as singular noun is tolerated today, may I suggest the more common plural usage.
Done

**Another few suggestions regarding the text are listed here:**
**page 18: the variable "swh_rejection_flag" formalised in the meta-description table at page 19, is called "swh_rejection_flags" at page 18. Please remove the inconsistency and check for other similar situations through the text.**
Done

**Page 23: ". . .ENVISAT sigma0, which seems to be stable with time. . ." Please try to add a couple of rows to justify this sentence, which looks too much qualitative.**
This comment only applies to the Product User Guide. The following statement has been added:
*as stated in Queffeulou et al. (2017)*

**Page 27: ". . .or could be matched to the L2P measurements for some reason." It may be interesting here to list briefly the main reasons considered for rejection.**
This paragraph has been removed since SSH measurements are not included in the Sea State CCI dataset.

**Page 45: the sentence "Besides, a test on swh rms (as provided in GDR for 1 Hz measurements) is performed, checking it is below an altimeter and swh dependant threshold" is actually unclear. Please rephrase and try to be more specific.**
This sentence has been rephrased in the Product User Guide as:
*Besides, a test on swh rms (as provided in GDR for 1 Hz measurements) is performed, based on the methodology proposed by Sepulveda et al., (2015). Measurements for which the swh rms is beyond a mission-dependent and swh-dependent threshold are classified as bad.*

**RC3**

**Additional comments**
**May I add a brief note about the paper. Actually, I have very few things to add respect to what I already said about the product user guide accompanying the dataset, and about the dataset itself.**
**It's a clear and well readable paper. Differently from the accompanying document, this paper looks almost immune from typos and grammar errors. My comments are substantially reflecting my impression in analysing the dataset and its accompanying doc, which I revised a few days ago.**

**In my opinion the explanations given in Section 3 are convincing, in particular I like the description of the de-noising method. Also, I find Fig. 9 very informative, but, you'll find just one question about it below (in "Technical points").**
**I still think that calibration/validation need further expansion (Section 4). Under this point of view, the science paper can adhere to the expansion suggested for the product user guide.**

**Also, for what regards Section 5, a further discussion of the regional trends identified in Sect. 5.2 may be helpful to many potential users.**
Section 5.2 has been further elaborated and regional differences in the spectral slopes are now clearly identified thanks to the computation of averaged-spectra in high, medium and low-vorticity regions (see also our response to RC2 from Referee #2). The regional differences are further discussed in this Section.

**Here I have some technical points**
**- Page 2, lines 12-15: please provide either two lines of explanation or cite a reference to justify the effect on sea state.**
The following references have been added here:
*Thomson, J., Rogers, W.E., 2014. Swell and sea in the emerging Arctic Ocean. Geophysical Research Letters 41, 3136–3140. https://doi.org/10.1002/2014GL059983*

*Idier, D., Bertin, X., Thompson, P., Pickering, M.D., 2019. Interactions Between Mean Sea Level, Tide, Surge, Waves and Flooding: Mechanisms and Contributions to Sea Level Variations at the Coast. Surv Geophys 40, 1603–1630. https://doi.org/10.1007/s10712-019-09549-5*

*Reguero, B.G., Losada, I.J., Méndez, F.J., 2019. A recent increase in global wave power as a consequence of oceanic warming. Nature Communications 10. https://doi.org/10.1038/s41467-018-08066-0*

**- Page 5, lines 17-18: as I suggested in the previous note, characterization of SWH for buoys < 200km can be interesting for many users.**
Validation results have been computed at various distances to the coast, both from in-situ observations (as depicted in Figure 1) and from model results. The results are now shown on Table 5 and described in Section 4.

**- Page 14: I'm not convinced by the x-axis notation (wavenumbers) in Fig. 9: does it span 1000 to 20 cycles/km going left-to-right? Seems contradicting the bandwidth reduction for higher order modes.**
The x-axis label was indeed confusing. It has been replaced by *wavenumber (km)*, which is in accordance with the x-axis values.

**- Page 15, line 23: is the subject of the sentence "It highlights..." still "Figure 9", mentioned 5 lines earlier? Please put an explicit subject.**
The sentence has been modified as follows:

*Figure 9, that shows how similar is the filter bank for pure noise and for SWH signal, therefore highlights the practical rule used for denoising, that compares the signal modulation in each IMF with the noise energy expected for the IMF of same rank.*

**- Pages 17-18: please note that legends and axes' texts are hard to read in Figures 10 and 12.**
Most figures of the manuscript have been reworked in order to increase the font size and improve their clarity.

**- Page 18 lines 13-17: If you calculate and show the differences between CCI and the dataset by Ribal and Young, you make the reader curious at least to get some conjecture about reasons to justify the differences. I suggest at least to specify the main differences (e.g. filtering) there.**
Section 5.1 has been further expanded (see RC2 from referee #2) and possible explanations for the observed differences are now provided.

**- Page 19 line 25: "...several locations worldwide which" –> "...several locations world-wide, which..."**
Corrected

**- Page 21: Fig. 14 is particularly small, especially the left pane individuating the regions of interest.**
This figure has been enlarged and modified to take into account both yours and Referee #2's comments.

**- Page 21 line 3: "...dataset exit..." –> "...dataset exists..."**
Corrected

**- Page 21 line 5: "...buoy datasets which include..." –> "...buoy datasets that include...**
Corrected

**Referees' comments**
Our response
*Change in the revised manuscript*

**Response to Referee Comment 2 (from Referee #2)**

**The authors describe a new sea-state dataset, for which they cross-calibrated ten satellite radar altimeters. They do a rigorous validation of the resulting dataset against another altimetry-derived dataset, in-situ data and wave models/reanalysis. Then the applications of the dataset are described and finally a list of shortcomings is provided.**

Major comments
**I think the structure is clear and the manuscript is easy to follow. The results are quite convincing, because of the extensive validation. However, I have the feeling that sections 4 and 5 lack depth. In most cases an observation is made, but the details or a possible explanation are never provided. It therefore gives the manuscript the character of a technical report. Therefore I have to recommend a major revision.**
We are thankful to Referee #2 for their constructive comments. As suggested, Sections 4 and 5 have been further elaborated in order to provide additional results and a more comprehensive analysis of these results. In particular, Section 4 now includes coastal and regional assessments of the CCI dataset, based on buoy and model comparisons (summarized in Table 5). The differences in the CCI dataset and the buoy and model outputs are discussed based on technical and geophysical considerations. Section 5 further discusses the differences in the climatology obtained from different altimeter and model products, and includes a new subsection dedicated to the analysis of long-term trends computed from these products. All changes in the manuscript can be found in the track-changes manuscript provided at the end of this document. We believe that these changes and the other ones specified below have clearly improved the manuscript and we hope that you will find it suitable for publication in the Earth System Science Data journal.

**Section 5 holds a discussion of applications, but there is only a comparison with other sea-state datasets and a hind that the geostrophic vorticity is connected to wave spectra. This should be expanded and rewritten.**
Section 5 has been expanded as detailed hereinbelow.

**Section 6.2 discusses the implementations of new retracking algorithms, but does not really provide the potential for near-future improvement. It relies largely on the delay/Doppler, but this only helps CryoSat-2, Sentinel-3 and Sentinel-6, which is only from 2010 onwards. Whatever retracking algorithm you implement, you will only find limited improvement in sea state near the coast.**
New retracking algorithms have been developed for both LRM and DDA altimeters, and many especially designed to bring advances in the coastal zone. To make these two points clearer in the text, we have made two changes:

i) p.22 l.10 We changed

*There is now a strong demand to improve the quality of altimetric wave height data through improved retracking methods*

to

*There is now a strong demand to improve the quality of altimetric wave height data from both LRM and DDA instruments through improved retracking methods*

ii) p.22 l.19-21  We changed

*The procedure (Schlembach et al.) involves comparison with external datasets (buoys and models), internal analysis of outlier rejection, quality flags, precision and spectral properties. The statistics are assessed both for different distances from the coast and varying values of SWH.*

to

*The procedure involves comparison with external datasets (buoys and models), internal analysis of outlier rejection, quality flags, precision and spectral properties. The results of this study show that a number of specially designed algorithms can deliver improved SWH retrieval in both open ocean and close to the coast, and for a range of sea state conditions (Schlembach et al., 2020).  The gains are achieved both through design of the retracking algorithms e.g. to avoid spurious signals in the tail of the waveform, and also through enhanced data selection using a data quality flag tuned to that specific retracker.*

**General comments**

**Make sure that in the captions clear information is provided. For example: in figure 10 a 'wave model hindcast' is used, but it is not directly clear which model is used here.**

This figure has been modified and now shows error metrics computed against in-situ measurements, instead of model outputs. Indeed, these results appear to be more adequate to discuss the impact of the calibration procedure on the results. Note that the comparisons against the model outputs are still displayed on Figure 11, which has been improved in order to show the impact of the denoising method on the Nortmalized Root Mean Squared Error.

**Minor comments**

**Page 2.**

**Line 1: As this is not an oceanography journal, remind the reader what is meant with sea state.**

The sentence has been changed as follows:

*Sea state, i.e. the description of wind sea and swell conditions at sea in terms of spectral or bulk wave parameters, is a key component of the coupling between the ocean and the atmosphere, the coasts and the sea ice.*

**Line 14: I do not see how an acceleration in sea level would translate into changes in sea state, except maybe for coastal waters. A line of explanation or a reference needed.**

Indeed, coastal water is implied here. The following reference has been added to make this clear:

Idier, D., Bertin, X., Thompson, P., Pickering, M.D., 2019. Interactions Between Mean Sea Level, Tide, Surge, Waves and Flooding: Mechanisms and Contributions to Sea Level

Variations at the Coast. Surv Geophys 40, 1603–1630.
https://doi.org/10.1007/s10712-019-09549-5

**Line 30: Remove the dot between cm and year.**
Corrected

**Page 5.**
**Line 4-19: The time series are filtered with a 1-hour filter, but most time series have a sampling rate of an hour of more. Should it be interpolated? Be also aware that in coastal zones an hour is quite long, so the interpolation/filtering method is quite important.**
There was a mistake as we actually used a 2-hour filter instead of a 1-hour filter. The buoy data were indeed interpolated on the satellite overpass time in order to take into account the possibly rapidly changing conditions in the coastal zone. This paragraph has been modified accordingly.

**Line 18: I think the validation should be shown for both, because you mention in the introduction that sea state is important for coastal processes.**
Validation results have been computed for various coastal strips (0km-50km, 50km-100km, 100km-200km, >200km), both from in-situ observations and from model results. These results highlight the lower quality of altimeter data near the coast. These new results are summarized in Table 5 and discussed in Section 4.

**Page 8.**
**Line 25: The figures seem quite convincing, but I wonder if there is latitudinal dependence remaining in the SWH differences, or even a water-depth dependence. The CCI sea-surface height products are corrected for a latitudinal dependence, which does not necessarily have to be orbit as we also observe it between Jason-1&2.**
Yes, there is a latitudinal dependence of the SWH difference, which is mostly due to the fact that the SWH errors depend on SWH and SWH is higher at high latitudes.

**Line 30: Are you considering only LRM data from CryoSat-2? If not, how is the SAR mode data processed: as Pseudo LRM or delay/Doppler?**
Only LRM measurements have been processed for CryoSat-2. This is now indicated in Section 3.2.

**Page 11.**
**Table 3: I would split TOPEX-A&B, because the instruments show clearly different properties. For TOPEX-A: is the cal1 applied or not and is it consistent with TOPEX-B?**
The distinction between Topex - Side A and Side B is now mentioned in the table together with the cycle number. As indicated, different calibration factors are applied to Topex-A and Topex-B.

**Page 12.**

**Line 23: "The IMFs ... algorithm." I think there is something missing in this sentence, maybe: "If the IMFs .. algorithm."**

The meaning looks correct to the authors but the sentence has been slightly modified to make it more explicit:

*The IMFs are calculated successively, the first one containing the shortest scales and the last one containing a trend, by construction of the algorithm.*

**Page 14.**
**Line 13: For white noise.**
Corrected

**Page 16.**
**Line 2: Why is there a negative bias for ERS, I would expect that the biases are removed by the cross-calibration? There is also a positive bias for the other missions, is this a problem in the data or the model?**

The calibration formulas presented here were derived during the GlobWave project based on a different reference dataset than the one used here for validation. The reasons why only ERS-1 presents a negative bias when compared against in-situ data remains unexplained. Figure 10 indicates that the GlobWave calibration for Jason-2, against which Jason-1, Saral, Cryosat-2 and Jason-3 are inter-calibrated, may have introduced a positive bias in these missions. This is now clearly stated in this Section, and a new section (Section 6.2) on the future improvements of the altimeter calibration methodology has been added.

**Page 17.**
**Merge section 4.1 and 4.2, it is very short.**

These sections have been expanded and Figure 10 has been improved in order to highlight the differences between the raw, calibrated and denoised SWH in terms of missions cross-consistency.

**Page 18.**
**Figure 12: A bias between TOPEX and Jason-1 appears to be introduced in the lower figure.**

As explained above, the inter-calibration method developed to merge the GlobWave-corrected missions and the most recent missions may have introduced some biases for specific missions. Moreover, the in-situ dataset used to calibrate the altimeter missions during GlobWave is not the same as the one used here for the validation, which may also explain some of the observed differences. This is now explained in Section 4.

**Figure 12: Maybe I missed it, but I am not sure it is clear how the ERS series is de-biased with respect to the Jason series. Maybe this can be added to table 3.**
See our answer to previous comments.

**Line 13-18: A bit more in-depth discussion is required why the differences are there. Is it related to the filtering or the de-biasing for example? The reader should know which dataset they should use when they have a certain application in mind. Also the**

**number of 2.5% looks quite nice, but as both datasets are coming from the same dataset I am not sure this is small.**

Section 5.1 on the global climatology inferred from different products has been expanded, and the observed differences are further explained based on the specificites of each product. The 2.5% global-mean difference between the CCI dataset and the one of Ribal and Young (2019) is only given here for indication. The spatial distribution of the differences shown on Figure 13 is a much better indication of the differences between these dataset.

**Page 19.**

**Line 5: I have the feeling that an issue in ENSO, PDO and IOD modeling are affecting the climatology. The authors have a strong oceanography background in the team. It should be possible to speculate at least what is causing this.**

The following explanation is now provided:

*In these results, the comparisons indicate that, even though ERA5 assimilates altimeter data, the ERA5 climatological mean SWH is substantially lower than CCI almost everywhere, except the eastern tropical Pacific and south tropical Atlantic where ERA5 clearly overestimates the wave climate. Once again, as for the comparison against Ribal & Young(2019), strong signatures are observed either side of the Equator. These are likely attributable to at least two factors. Firstly, ERA5 generally under-estimates SWH in stormy areas, except in the deep tropics where the wave climate is dominated by long period swell. Recent changes have been made to the ERA5 wave physics package to try to solve some of these issues. Secondly, in the tropical Pacific Ocean, the impact of the equatorial and counter-equatorial currents are clearly visible. This corresponds to the absence of ocean surface currents, both in the atmosphere boundary layer and in the wave model component of ERA5. It is also affected by the relatively coarse (32 km) wind fields, which lead to loss of information in the wave model.*

**Section 5.1: What I am missing here is a comparison of geographical trends and a comparison of the variability. As mentioned in the introduction, changes in sea state are important for a variety of reasons, but the change is completely left out of the discussion. It will probably also give you more insight into the previous comment.**

A new subsection (5.2) on the long-term trends derived from altimeter products and model reanalysis, and a new figure (Fig 14) showing the global distribution of annual mean SWH trend estimates over 1992-2017 for Ribal & Young (2019), the CCI dataset, and two ECMWF reanalysis have been added. This analysis is based on the results of Timmermans et al. (2020).

Timmermans, B.W., Gommenginger, C.P., Dodet, G., Bidlot, J.-R., 2020. Global Wave Height Trends and Variability from New Multimission Satellite Altimeter Products, Reanalyses, and Wave Buoys. Geophysical Research Letters 47, e2019GL086880. https://doi.org/10.1029/2019GL086880

**Line 10: For illustration purposes.**

Corrected

**Section 5.2: This section suggests that the wave energy is higher in high-geostropic-vorticity regions. Instead of showing a couple of examples, I would compute an average spectrum over low-, medium- and high-vorticity regions. The authors show a correlation between the two quantities, but do not discuss how both are affecting each other. I am not sure what I should get from this section, it should be further elaborated. The topic also seems a bit selective.**

Following your suggestions we have computed averaged spectrum over low-, medium- and high-vorticity regions. For each vorticity level we have selected two regions in order to assess the consistency of the results. Spectral analysis within each of these regions indicate steeper spectral slope (~$k^{-2.5}$) for the high vorticity regions in comparisons to the lower vorticity regions (~$k^{-1.5}$), as a result of wave-current interactions. While the detailed investigations of these spectral shapes is left to future studies, the aim of this section was to highlight the benefit of the denoised SWH provided in the CCI dataset in order to study SWH variability at scales lower than 100km. This is now more clearly stated in this Section.

**Page 21. Figure 14: The figure for vorticity is too small.**
The figure has been enlarged.

**Line 13: Explain fading noise.**
To explain it, the original sentence has been expanded into three:
*For conventional (low rate mode) altimetry, all the information on SWH is encrypted in the few bins on the leading edge of the waveform (Figure 15a). The actual echo observed at any bin is the sum of the contributions from many incoherent reflecting points on the sea surface; the effect of this "fading noise" is that the power recorded in the mean waveform has an intrinsic variability that will have a strong effect on parameters calculated from only a few waveform bins. Also, in the coastal zone, unwanted reflections from nearby land or sheltered bays (Gomez-Enri et al., 2010); can affect the quantity and quality of SWH estimations within 20 km of the coast (Passaro et al., 2015).*

**Line 15: Quantity and quality.**
Corrected

**Line 13-15: One of the major issues is the change of wave shape in shallow areas, which is not discussed here. This is not modeled in standard retrackers.**
This is now stated as follows:
*Also, in the coastal zone, unwanted reflections from nearby land or sheltered bays (Gomez-Enri et al. 2010) and changes in the wave shape due to wave-bottom and wave-current interactions (Ardhuin et al., 2012) can affect the quantity and quality of SWH estimations within 20 km of the coast (Passaro et al., 2015).*

**Line 22: Multilooking is already applied to reduce speckle in LRM waveforms. Typically 90 waveforms are averaged. The great benefit of delay/Doppler is the enhanced along-track resolution, which allows you to get closer to the coast under certain angle-of-attacks. On top of that it increases the number of independent looks per second by a factor of +/-2, which can be used to improve precision (reduce speckle).**

Our original text was confusing, and has been changed.
*multi-looking*
has been replaced by

[revised manuscript text omitted]

$^{(1)} dh = -0.0685 + 6.0426.10^{-4} cycle + 7.7894.10^{-6} cycle^2 - 6.9624.10^{-8} cycle^3$

[revised manuscript text omitted]